# Glycolysis inhibition in tuberculosis-driven metabolic rewiring reduces HIV-1 spread in macrophages

Zoï Vahlas[1,2], Clara Deyts[1,2], Steven Fried[6], Myriam Ben Neji[1], Maxime Pingret[1], Natacha Faivre[1], Sarah C Monard[1,2], Quentin Hertel[3], Mariano Maio[2,4], Joaquina Barros[2,4,5], Alexandre Lucas[6], Thien Phong Vu Manh[7], Marcelo Corti[8], Renaud Poincloux[1,2], Fabien Blanchet[3], Brigitte Raynaud-Messina[1,2], Fabien Letisse[1], Olivier Neyrolles[1,2], Geanncarlo Lugo-Villarino[1,2,*], Luciana Balboa[2,4,5,*], Christel Vérollet[1,2,*]

**Tuberculosis (TB) is a significant aggravating factor in individuals living with HIV-1, the causative agent for AIDS. Both *Mycobacterium tuberculosis* (Mtb), the bacterium responsible for TB, and HIV-1 target macrophages. Understanding how Mtb subverts these cells may facilitate the identification of new druggable targets. Here, we explored how TB can induce macrophages to form tunneling nanotubes (TNT), promoting HIV-1 spread. We found that TB triggers metabolic rewiring of macrophages, increasing their glycolytic ATP production. Using several pharmacological inhibitors, glucose deprivation, and glucose or galactose supplementation, we discovered that disrupting aerobic glycolysis significantly reduces HIV-1 infection in these macrophages. Glycolysis is essential for tunneling nanotubes formation, which facilitates viral transfer and cell-to-cell fusion. Importantly, HIF-1α activation contributes to these processes. Overall, these data might facilitate the development of targeted therapies aimed at inhibiting HIF-1α–dependent glycolytic activity in TB-induced immunomodulatory macrophages to ultimately halt HIV-1 dissemination in coinfected patients.**

## Introduction

Tuberculosis (TB) and AIDS are among the deadliest diseases caused by single infectious agents. A significant issue in the AIDS epidemic is the synergy between the HIV-1 and *Mycobacterium tuberculosis* (Mtb), the etiological agents of AIDS and TB, respectively. Globally, Mtb is the most frequent coinfection in patients with HIV-1 and represents a major risk factor for increased morbidity and mortality (WHO Global Tuberculosis Report 2021). Addressing the TB challenge within the AIDS epidemic requires a comprehensive understanding of the pathophysiology of HIV/Mtb coinfection, including the role of immunometabolism (Esmail et al, 2018).

Despite CD4[+] T lymphocytes being the primary target for HIV-1, macrophages provide a crucial niche for both pathogens, allowing Mtb to evade immune responses and HIV-1 to persist and replicate, thereby exacerbating the impact of coinfection (Mayer-Barber & Barber, 2015; Tan & Russell, 2015; Cohen et al, 2018). On the one hand, lung macrophages are the primary target of Mtb because they are the primary immune cells responsible for engulfing and attempting to eliminate pathogens through phagocytosis, thereby providing an intracellular environment that the bacteria can exploit to survive and replicate. On the other hand, infected macrophages are found in tissues of HIV[+] patients and simian immunodeficiency virus (SIV)–infected nonhuman primates (NHPs), playing an important role in the pathogenesis (Honeycutt et al, 2016, 2017; Rodrigues et al, 2017). In addition, multiple studies demonstrate that tissue macrophages, including microglia and urethral, gut, and lung macrophages, can serve as reservoirs for HIV-1 in patients undergoing antiretroviral therapy (Ganor et al, 2013, 2019; Sattentau & Stevenson, 2016). In the lung, for example, the macrophage compartment is the main target of HIV-1 (Jambo et al, 2014; Avalos et al, 2016; Schiff et al, 2021). Our team showed recently that the abundance of lung macrophages becomes augmented in NHPs with active TB and exacerbated in those coinfected with SIV, acquiring an immunomodulatory phenotype distinguished by the overactivation of the interleukin-10 (IL-10)/signal transducer and activator of transcription 3 (STAT3) axis (Souriant et al, 2019; Dupont et al, 2020, 2022). This phenotype is closely related to the so-called "M(IL-10)" activation program (Murray et al, 2014), which is abundantly found in the

[1]Institut de Pharmacologie et de Biologie Structurale (IPBS), Université de Toulouse, CNRS, Toulouse, France  [2]International Research Project CNRS "MAC-TB/HIV", Toulouse, France  [3]Institut de Recherche en Infectiologie de Montpellier (IRIM), Université de Montpellier, CNRS UMR9004, Montpellier, France  [4]Instituto de Medicina Experimental (IMEX)-CONICET, Academia Nacional de Medicina, Buenos Aires, Argentina  [5]Instituto de Investigaciones Biomédicas en Retrovirus y Sida (INBIRS)-CONICET, Universidad de Buenos Aires, Buenos Aires, Argentina  [6]Institut des Maladies Métaboliques et Cardiovasculaires, Inserm, Toulouse, France  [7]Aix-Marseille University, CNRS, INSERM, CIML, Centre d'Immunologie de Marseille-Luminy, Turing Center for Living Systems, Marseille, France  [8]Hospital de Infecciosas Dr. F.J. Muñiz, Buenos Aires, Argentina

Correspondence: Geanncarlo.Lugo@ipbs.fr; christel.verollet@ipbs.fr
*Geanncarlo Lugo-Villarino, Luciana Balboa, and Christel Vérollet contributed equally to this work

pleural cavity of patients with active TB and reproduced in vitro by exposing human monocytes to TB-associated microenvironments (Lastrucci et al, 2015). Further work with these TB-induced immunomodulatory macrophages demonstrated an increased susceptibility to HIV-1 replication and spread via the formation of tunneling nanotubes (TNTs), which facilitate the transfer of the virus between macrophages, and trigger their fusion, leading to the formation of highly virus-productive multinucleated giant cells (MGCs) (Souriant et al, 2019). Of note, MGCs are considered the pathological hallmarks of HIV-1 infection in macrophages (Orenstein, 2000; Verollet et al, 2010, 2015). TNTs are membranous channels containing F-actin that connect cells over long distances and can be hijacked by pathogens to circumvent the immune system (Dupont et al, 2018; Zurzolo, 2021). However, the molecular mechanisms linking TB infection to TNT formation, stability, and function in immunomodulatory macrophages are not well understood, and understanding them might contribute to the development of novel targeted therapies.

Chronic host–pathogen interactions in TB result in extensive metabolic remodeling in both the host and the pathogen (Huang et al, 2019; Kumar et al, 2019; Llibre et al, 2021). In fact, the success of Mtb as a pathogen largely relies on its ability to adapt to the intracellular milieu of macrophages and exploit their metabolic activity. In chronic infectious diseases, there is often a shift in the macrophage activation program from the M1 phenotype, which primarily relies on aerobic glycolysis, toward the M2 phenotype, which depends heavily on oxidative phosphorylation (OXPHOS), at the site of inflammation (Lugo-Villarino et al, 2011; Biswas & Mantovani, 2012; Saha et al, 2017). This metabolic rewiring is associated with the adaptive immune transition from acute to chronic phases (Gleeson et al, 2016; Cumming et al, 2018; Marin Franco et al, 2020; Vrieling et al, 2020; O'Maoldomhnaigh et al, 2021). In general, macrophages infected by live Mtb acquire the M1 phenotype, characterized by elevated production of pro-inflammatory molecules. They rely on aerobic glycolysis and the pentose phosphate pathway to meet their bioenergetic and metabolic requirements. However, results can vary depending on differences in Mtb strains, multiplicity of infection, macrophage origins, and measurement time points (Gleeson et al, 2016; Cumming et al, 2018; Marin Franco et al, 2020; Vrieling et al, 2020). In such M1 macrophages, OXPHOS and fatty acid oxidation are dampened. Nonetheless, the untimely overproduction of lactate, the end-product of aerobic glycolysis, disrupts macrophage metabolism, leading to an attenuated glycolytic shift upon subsequent stimulation with irradiated Mtb and reduced pro-inflammatory cytokine production (O'Maoldomhnaigh et al, 2021). Also, a shift from glycolysis to OXPHOS is observed in M1 macrophages when they are exposed to the acellular fraction of pleural effusions (PE) from TB patients (TB-PE), which is considered hereafter a genuine TB-associated microenvironment (Marin Franco et al, 2020). Of note, human monocyte differentiation under TB-PE yields immunoregulatory macrophages (Lastrucci et al, 2015). Different metabolic pathways are prevalent in macrophages depending on their ontogeny, the state of TB (active or latent), and the virulence of the pathogen (live or irradiated Mtb); these states can be remarkably reversible depending on environmental cues (Pandey & Sassetti, 2008; de Carvalho et al, 2010; Beste et al, 2013;

Zimmermann et al, 2017; Cano-Muniz et al, 2018; Cumming et al, 2018; Llibre et al, 2021). Therefore, results about how Mtb impacts macrophage metabolism are conflicting.

Although the role of macrophage metabolism is well described in TB, little is known for HIV-1 host cells (Shehata et al, 2017; Saez-Cirion & Sereti, 2021). Our understanding of how metabolism affects HIV-1 infection primarily comes from studies in CD4 T lymphocytes (Loisel-Meyer et al, 2012; Craveiro et al, 2013; Clerc et al, 2019; Valle-Casuso et al, 2019; Saez-Cirion & Sereti, 2021). To be efficiently infected by HIV-1, these cells must be metabolically active. In this regard, increases in both aerobic glycolysis and OXPHOS are crucial for the early steps of HIV-1 infection (Clerc et al, 2019; Valle-Casuso et al, 2019). In macrophages, HIV-1 has been shown to alter their metabolic status (Hollenbaugh et al, 2011; Castellano et al, 2019; Saez-Cirion & Sereti, 2021). A recent report indicated that a metabolic shift toward aerobic glycolysis can reactivate HIV-1 replication in macrophages (Real et al, 2022). However, the underlying mechanisms and consequences remain unknown. Although the study of immunometabolism on HIV-1 infection and progression is an emerging field, very little research has focused on macrophage metabolism, especially in the context of coinfection with Mtb.

In summary, our research demonstrates that TB significantly exacerbates HIV-1 infection by inducing metabolic reprogramming in macrophages, leading to increased glycolytic ATP production. This metabolic shift promotes the formation of TNTs, which facilitate viral transfer and cell-to-cell fusion. The study highlights the critical role of glycolysis in TNT formation and the subsequent expression of the sialoadhesin Siglec-1, which enhances HIV-1 binding and TNT stabilization. Importantly, inhibiting glycolysis significantly reduces HIV-1 exacerbation in TB-infected macrophages, suggesting that targeting glycolytic pathways could be a promising therapeutic strategy to prevent HIV-1 dissemination in coinfected patients.

## Results

### TB-associated microenvironment increases glycolysis in macrophages

Understanding the TB-associated microenvironment in the context of HIV-1 coinfection is essential because TB significantly worsens the prognosis for individuals infected with HIV-1. Although it is rare to find cells coinfected with both pathogens, studying these microenvironments is vital, as TB and HIV-1 often occupy adjacent niches, such as the lungs, leading to enhanced viral dissemination and immune system manipulation. To determine whether the metabolic profile of macrophages is modulated in TB-associated microenvironments, we employed a previously described in vitro model (Lastrucci et al, 2015; Souriant et al, 2019; Dupont et al, 2020, 2022), which consists of differentiating primary human monocytes into macrophages in the presence of a TB-associated microenvironment, such as conditioned medium generated from Mtb-infected macrophages (cmMTB). These macrophages display an immunomodulatory phenotype characterized by a CD16$^{+-}$ CD163$^+$MerTK$^+$PD-L1$^+$ receptor signature, nuclear translocation

of phosphorylated STAT3, and functional properties like protease-dependent motility, suppression of T-cell activation, and susceptibility to Mtb or HIV-1 infection (Lastrucci et al, 2015; Souriant et al, 2019; Dupont et al, 2020, 2022). Conditioned medium of noninfected macrophages (cmCTR) was used as a control. The overall energy phenotype was evaluated using the Seahorse technology at day 3 of monocyte differentiation into macrophages with either cmCTR or cmMTB, or with nothing (RPMI). First, basal oxygen consumption rate (OCR) was plotted against extracellular acidification rate (ECAR) using the Seahorse Mito Stress assay. The resulting metabolic phenogram revealed that cmMTB treatment shifts the energy metabolism of the cells toward a more energetic and glycolytic profile compared with cmCTR-treated cells or control cells (RPMI only) (Fig S1A). Thus, cmCTR was used as the control condition in the next experiments.

Then, the intracellular ATP production rate was measured in these cells using the Seahorse ATP Rate assay. We found that total ATP production was strongly increased in cmMTB-differentiated macrophages compared with cmCTR-treated cells or control cells (Fig 1A). The measurements of basal extracellular acidification rate (ECAR) and basal oxygen consumption rate (OCR) were used to calculate ATP production rate from glycolysis (GlycoATP) and mitochondrial OXPHOS (MitoATP), respectively, at different time points (Figs 1 and S1B and C). We found that the intracellular GlycoATP production was significantly increased in cmMTB-differentiated macrophages compared with control cells (Fig 1B). Kinetic analyses revealed that the increase in GlycoATP production began as early as day 1 and peaked at day 3 of monocyte differentiation under cmMTB (Fig S1B). However, this change in the metabolic state was transitory, as the difference in GlycoATP production observed between cmCTR and cmMTB conditions vanished by day 6 (Fig S1B).

Importantly, high ATP production, especially GlycoATP production, was also observed during monocyte differentiation into immunomodulatory macrophages in another TB-associated microenvironment, such as the acellular fraction of TB-PE. This contrasted with conditioning with PE obtained from patients with heart failure (HF-PE), which served as the control (Fig 1C and D). Simultaneously, the level of MitoATP also increased in cmMTB- or TB-PE–differentiated macrophages compared with their control counterparts (Fig S1C and D), displaying kinetics similar to those of GlycoATP production.

As measured by the Seahorse Mito Stress assay, we found that the maximal respiration and spare respiratory capacity were slightly increased in cmMTB-differentiated macrophages relative to those conditioned with cmCTR (Fig S1E and F), as measured by the Seahorse Mito Stress assay. Because MitoATP is different between cmMTB- and cmCTR-differentiating macrophages, the number of mitochondria was assessed using transmission electron microscopy (TEM) (Fig S2A and B), and mitochondrial biomass was measured by flow cytometry (Fig S2C); both parameters were found to be comparable between the two conditions. Using the MitoSOX fluorescent probes to measure intracellular superoxide formation, no statistical difference in oxidative stress was observed between these two cell populations (Fig S2D). This suggests that the increase in maximal respiration and spare respiratory capacity is not due to cellular stress.

Next, to understand the relative contributions of mitochondrial respiration and glycolysis to the bioenergetic profile of macrophages differentiated under TB-associated microenvironments, we compared the relative use of mitochondrial versus glycolytic pathways for ATP production between cmMTB- and cmCTR-differentiated macrophages at day 3. Approximately 90% of ATP production in macrophages differentiated with cmCTR originated from OXPHOS; this parameter decreased to 70% when conditioned with cmMTB (Fig 1E and F). Consistently, the percentage of GlycoATP increased from 10% to more than 25% in cmMTB-differentiated macrophages compared with control cells (Fig 1E and G), leading to an overall decrease in the Mito/GlycoATP ratio (Fig 1H). Similar results were obtained with TB-PE (Fig 1I–L), demonstrating increased glycolytic activity in immunomodulatory macrophages differentiated under different TB-associated microenvironments.

To further our analysis of the glycolytic activity of cmMTB-differentiated macrophages, the Seahorse Glycolytic Rate assay was used. As shown in Fig 2A, all the parameters related to glycolysis (basal, % of proton efflux rate [PER], and compensatory) were significantly increased in cmMTB-differentiated cells compared with controls (Fig 2A). The shift to aerobic glycolysis by TB was further supported by a significant enrichment of glycolytic genes observed in cmMTB-differentiated macrophages compared with control cells (Fig 2B). This finding was obtained by reanalyzing our previously published genome-wide transcriptomic data (GEO submission GSE139511) using a GSEA-based approach (Dupont et al, 2022). In contrast, genes of the OXPHOS pathway failed to be enriched in the cmMTB-differentiated macrophages (Fig S2E). To validate this in silico analysis, glucose uptake was measured by fluorescent d-glucose analog 2-[N-(7-nitrobenz-2-oxa-1,3-diazol-4-yl) amino]-2-deoxy-D-glucose (2-NBDG). The results indicate a significantly elevated uptake in cmMTB-differentiated macrophages compared with control cells (Fig 2C). This increase was reflected in the higher glucose consumption found in the extracellular medium of these macrophages, as measured dynamically using a nuclear magnetic resonance (NMR)–based metabolomic approach (Fig 2D). This approach also revealed that extracellular lactate production by cmMTB-differentiated macrophages was higher than in controls (Fig 2E). The elevated lactate release was also verified by classical spectrophotometric analysis in macrophages differentiated under cmMTB or TB-PE conditions, compared with those differentiated under cmCTR or HF-PE conditions, respectively (Fig 2F and G). Glycolytic reprogramming in macrophages is orchestrated by hypoxia-inducible factor-1 alpha (HIF-1$\alpha$), which increases the expression of glycolytic enzymes and pro-inflammatory cytokines (Palsson-McDermott et al, 2015; Gleeson & Sheedy, 2016). Thus, HIF-1$\alpha$ expression was examined in our experimental conditions. Accordingly, although the protein expression level of HIF-1$\alpha$ was not modified between the two cell populations (Fig 2H), only cmMTB-differentiated macrophages exhibited its frequent translocation from the cytoplasm to the nucleus (Fig 2I and J), suggesting that TB-associated microenvironments favor glycolysis through HIF-1$\alpha$ activation.

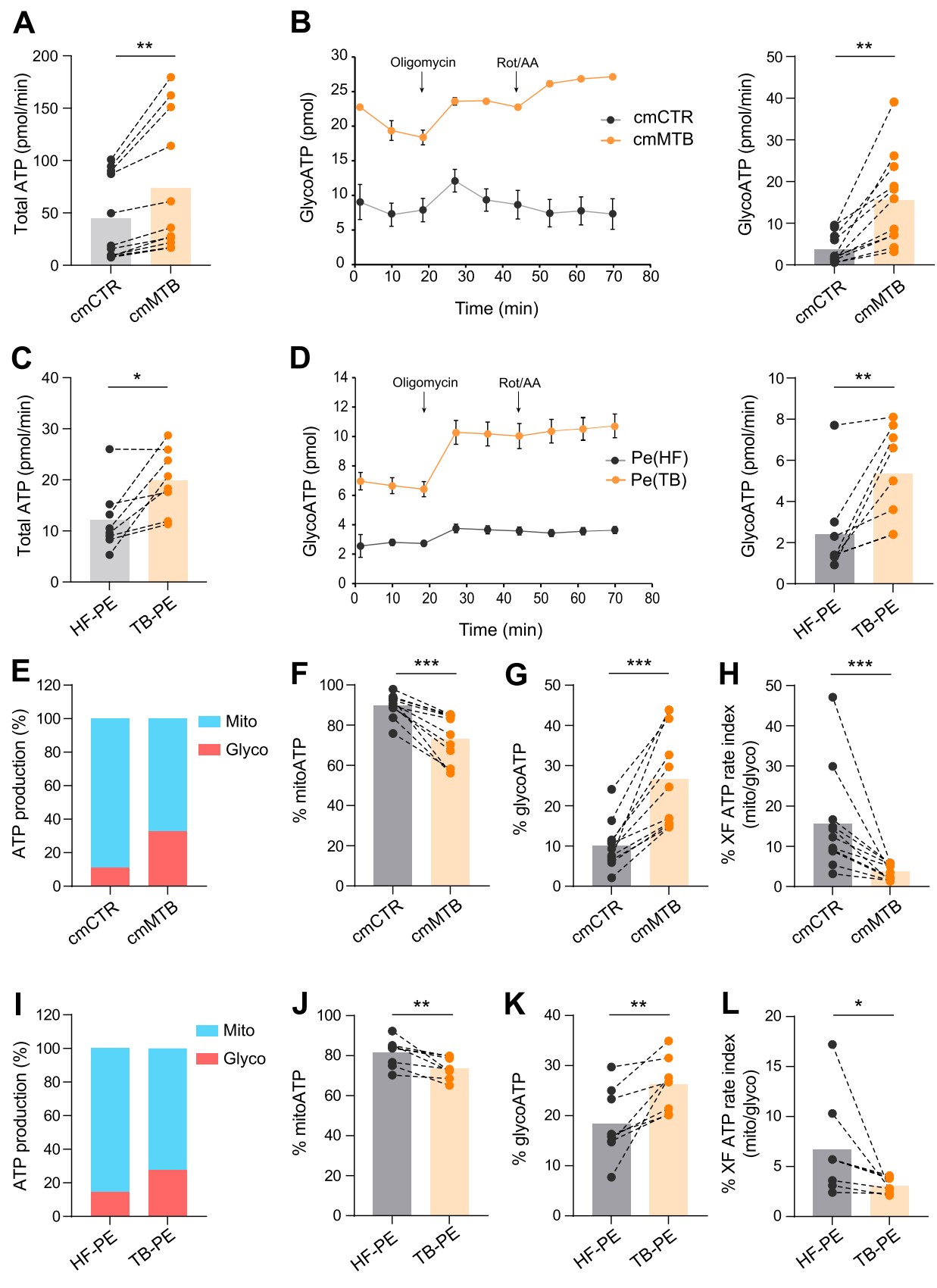

## Glycolysis induced by TB-associated microenvironments exacerbates HIV-1 infection of immunomodulatory macrophages

Because our immunomodulatory macrophage model is permissive to HIV-1 infection (Lastrucci et al, 2015; Souriant et al, 2019; Dupont et al, 2020, 2022), we investigated whether its elevated glycolytic activity contributes to this susceptibility. To achieve this, the metabolism of cmMTB-differentiated macrophages was modulated using a pharmacological approach: UK5099 was used to partially block pyruvate entry into mitochondria, thereby enhancing glycolysis. In addition, oxamate and GSK 2837808A were used to specifically target lactate dehydrogenase, the enzyme that converts pyruvate to lactate, thereby diminishing glycolysis. We also used the glucose analog 2-deoxy-d-glucose (2-DG) to block glucose uptake (Fig S3A). CmMTB-differentiated macrophages were incubated with the indicated drugs for 24 h before HIV-1 infection (Fig S3B). A VSVG-pseudotyped NLAD8 strain was used to enhance HIV-1 entry regardless of the presence of CD4/CCR5 entry receptors that might be affected by metabolic changes. First, the multiple drug treatments did not affect cell viability, as verified by flow cytometry (Fig S3D) and immunofluorescence (Figs 3A and S3C). As expected, inhibiting glycolysis with oxamate reduced lactate release from cmMTB-differentiated macrophages, whereas enhancing glycolysis with UK5099 slightly increased it (Fig 3B). Importantly, enhanced glycolysis significantly increased the number of HIV-1–infected cmMTB-differentiated macrophages, as evidenced by increased intracellular detection of the HIV-1-Gag protein by immunofluorescence (Fig 3A and C). In contrast, HIV-1 infection was significantly reduced when glycolysis was inhibited with oxamate or GSK 2837808A, and was slightly reduced with 2-DG (Figs 3C and S3E and G). These contrasting effects in HIV-1 infection through glycolysis modulation were also confirmed by flow cytometry analysis (Fig S3I and J). Regarding the formation of MGCs (Orenstein, 2000; Verollet et al, 2010, 2015), their numbers decreased or increased with diminished or enhanced glycolysis, respectively (Figs 3D and S3F). Of note, blocking glucose uptake with 2-DG completely abrogates HIV-1–infected cell fusion into MGCs (Figs 3D and S3G and H). In line with these results, we observed at least a 50% decrease in HIV-1 infection and MGC formation in monocytes differentiated with TB-PE under conditions of diminished glycolysis (Fig 3E–G).

To complement our pharmacological approach, we modified glycolysis by depriving the extracellular medium of glucose (Fig 3H–K). Accordingly, the lactate release by cmMTB-differentiated macrophages was reduced compared with the normal conditions (25 mM glucose) (Fig 3I). This deprivation also had a strong inhibitory effect on HIV-1 infection and MGC formation, without affecting cell density, compared with control cells (Figs 3H, J, and K). Interestingly, using galactose as the sole carbon

source in place of glucose, which generates ATP only via OXPHOS, did not restore HIV-1 infection or MGC formation (Fig S3K and L).

Collectively, these results demonstrate that a shift toward aerobic glycolysis in our immunomodulatory macrophage model enhances its susceptibility to HIV-1 infection and the formation of infected MGCs.

## TB-induced aerobic glycolysis promotes cell-to-cell dissemination of HIV-1 via TNT formation

Previous work demonstrated that TB-associated microenvironments alter HIV-1 dissemination via TNTs, which facilitate virus transfer between macrophages, trigger cell fusion, and lead to the formation of highly virus-productive MGCs. No other steps of the HIV-1 viral cycle were affected in these cells (Lastrucci et al, 2015; Souriant et al, 2019; Dupont et al, 2020, 2022). To determine whether glycolysis regulates the cell-to-cell transfer of HIV-1, a coculture was performed between uninfected recipient (labeled with CellTracker+) and HIV-1–infected donor (Gag+) cmMTB-differentiated macrophages for 24 h (Fig S4A). This time point allows sufficient donor macrophages to transfer the virus to recipient cells primarily through fusion and mainly in a TNT-dependent manner (Souriant et al, 2019). Accordingly, among HIV-1-Gag+ cells, about 35% were MGCs double-positive for Cell-Tracker (Fig 4A and B). Of note, infection by newly produced viruses is unlikely within 24 h. More importantly, both cell-to-cell transfer and fusion between these macrophages were significantly reduced when donor cells were treated with a glycolysis inhibitor before HIV-1 infection (Fig 4A and B, left). To further confirm that only the cell-to-cell transfer of the virus is affected by glycolysis, glycolysis was specifically inhibited at the time of coculture, corresponding to 3 d after HIV-1 infection of donor cells (Fig S4A). Under these conditions, the number of double-positive cells was also decreased (Fig 4B, right), suggesting that glycolysis controls this mode of viral transmission without affecting earlier steps of the viral cycle.

HIV-1 transfer between macrophages is usually mediated through TNTs, whereby HIV-1 can be found within the thicker, longer, and more stable versions of these structures, which are characterized by the presence of microtubules (Dupont et al, 2020). Using GFP-tagged virus and live imaging, thick TNTs containing HIV-1 were observed as a preliminary step to the fusion of infected macrophages (Fig S4B and Video 1). As glucose transfer is mainly mediated by the glucose transporter type 1 (GLUT-1), we investigated its localization related to TNTs in cmMTB-differentiated macrophages. Interestingly, we found that GLUT-1 accumulates at the tip of TNT in macrophages (Fig 4C). To further investigate the role of glycolysis in TNT formation, we modulated its activity in HIV-1–infected cmMTB-differentiated macrophages using our

**Figure 1. TB-associated microenvironments increase aerobic glycolysis in macrophages.**
Monocytes from healthy subjects were treated either with conditioned medium from mock- (cmCTR, gray) or Mtb-infected macrophages (cmMTB, orange), or with heart failure (HF-PE, gray) or pleural effusions (PE) from TB patients (TB-PE, orange) for 3 d and analyzed using Agilent Seahorse XFe24 Analyzer. **(A, C)** Dot plots showing total ATP production. **(B, D)** GlycoATP rate after addition of oligomycin and rotenone/antimycin A (ROT/AA) over time (left) and dot plots showing total GlycoATP production (right). **(E, I)** Percentages of MitoATP and GlycoATP production relative to overall ATP production, representative experiments. **(F, J)** Dot plots showing the percentages of MitoATP. **(G, K)** Dot plots showing the percentages of GlycoATP. **(H, L)** Scattered plots showing ATP rate index (% XF). Each circle within vertical plots represents a single donor. Histograms represent mean values. Statistical analysis: t test data with normal distribution; *$P \leq 0.05$; **$P \leq 0.01$; ***$P \leq 0.001$.

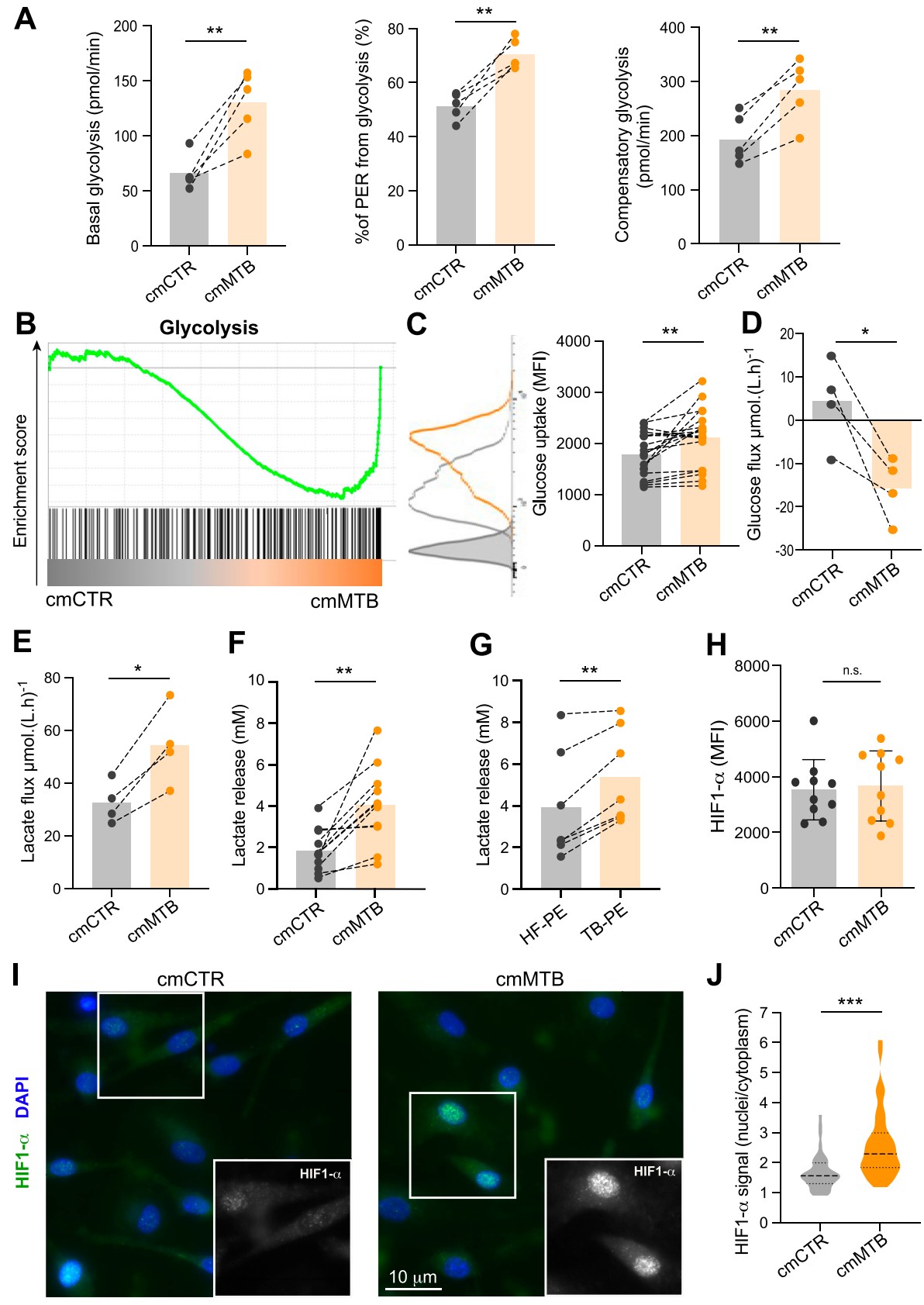

pharmacological approach. Scanning electron microscopy analyses revealed that enhanced glycolysis significantly increased the number of cells forming TNTs, whereas diminished glycolysis led to a deficiency of cells forming these structures (Fig 4D). These observations were confirmed through an IF quantification approach (Figs 4E and S4C). Moreover, measurement of the number of thick and thin (lacking microtubules) TNTs (Souriant et al, 2019; Dupont et al, 2020) illustrated the importance of the glycolytic activity of macrophages for their formation; that is, enhanced glycolysis increased both types of TNT formation, whereas diminished glycolysis decreased them (Fig S4D). Importantly, TNT formation was also inhibited by reducing glucose levels in the medium (Fig 4F), and under glycolysis inhibition using GSK and 2-DG treatments (Fig S4D–F). As Siglec-1, a type I lectin receptor that recognizes and binds sialic acid–containing glycoproteins, stabilizes thick TNTs and enhances HIV-1 binding and transfer (Dupont et al, 2020), we investigated whether glycolysis affected its expression in our immunomodulatory macrophage model. Flow cytometry analyses revealed that the surface expression of Siglec-1 was reduced under diminished glycolysis or glucose deprivation, whereas it was increased under enhanced glycolysis (Fig 4G and H). Finally, to consolidate the notion that TB-induced aerobic glycolysis exacerbates HIV-1 spread in macrophages via TNTs, these structures were pharmacologically inhibited using a TNT inhibitor (TNTi) (Hashimoto et al, 2016; Souriant et al, 2019). In cmMTB-differentiated macrophages with enhanced glycolysis, TNTi treatment significantly decreased both TNT formation and HIV-1 spread among cells (Fig 4I–K).

Altogether, these findings demonstrate that TB-induced aerobic glycolysis in immunomodulatory macrophages enhances TNT formation, thereby promoting HIV-1 spread among these cells.

### HIF-1α promotes cell-to-cell dissemination of HIV-1 via TNT

Given that TB-induced aerobic glycolysis enhances HIV-1 dissemination through TNT formation, we next investigated whether this process is regulated by the hypoxia-responsive transcription factor HIF-1α. Immunofluorescence analyses revealed that in HIV-1–infected macrophages, HIF-1α was predominantly translocated to the nucleus, particularly in cells that had undergone fusion and formed MGCs (Fig 5A). This nuclear enrichment of HIF-1α in HIV-Gag–positive MGCs suggests that HIF-1α is transcriptionally active in macrophages engaged in cell–cell fusion and TNT-mediated interactions.

To directly assess the role of HIF-1α, we performed siRNA-mediated knockdown in HIV-1–infected macrophages. Efficient silencing of HIF-1α was confirmed by two independent approaches (Fig S5): immunofluorescence microscopy demonstrated a significant decrease in the proportion of nuclei positive for HIF-1 (Fig S5A and B), and capillary Western blot analysis confirmed reduced (~60%) HIF-1α protein expression (Fig S5C).

Functionally, HIF-1α depletion led to a consistent reduction in HIV-1 cell-to-cell dissemination. Although the infection index showed a downward trend upon HIF-1α silencing (Fig 5B and C), the most pronounced effect was observed on macrophage fusion (Fig 5B and D), as measured by the frequency of multinucleated HIV-1–positive cells, in the absence of HIF-1α. In line with these findings, the proportion of cells forming TNTs was significantly reduced upon HIF-1α knockdown (Fig 5E), indicating that HIF-1α is required for efficient TNT biogenesis and/or stabilization. These effects are consistent with those observed upon pharmacological inhibition of glycolysis, supporting the notion that HIF-1α acts upstream of the metabolic program that fuels TNT formation.

Altogether, these data identify HIF-1α as a key regulator of TNT-dependent HIV-1 dissemination in macrophages. By promoting glycolysis, TNT formation, and macrophage fusion into MGCs, HIF-1α establishes a TB-driven metabolic and structural framework that facilitates efficient cell-to-cell spread of HIV-1 independently of classical viral replication steps.

## Discussion

Macrophages play a pivotal role in HIV-1 dissemination and the establishment of persistent viral reservoirs across various host tissues, including the lungs, where they are the primary target for Mtb (Sattentau & Stevenson, 2016; Rodrigues et al, 2017; Ganor et al, 2019; Hendricks et al, 2021). The detrimental synergy between Mtb and HIV-1 significantly impacts the host, necessitating a deeper understanding of how TB-associated microenvironments enhance HIV-1 infection dynamics (Esmail et al, 2018). Our recent research has identified a specific immunomodulatory macrophage phenotype induced by TB-associated microenvironments, which becomes highly susceptible to both Mtb and HIV-1 infections and replication.

**Figure 2. TB-associated microenvironments increase aerobic glycolysis in macrophages.**
**(A, B, C, D, E, F, G, H, I, J)** Monocytes were treated either with conditioned medium from mock-infected macrophages (cmCTR, gray) or Mtb-infected macrophages (cmMTB, orange), or with pleural effusions from heart failure (HF-PE, gray) or from TB (TB-PE, orange) patients. **(A)** Analysis of metabolic parameters using Agilent Seahorse XFe24 Analyzer (Glycolytic Rate assay): basal glycolysis, % of proton efflux rate (PER), and compensatory glycolysis were measured in cmCTR- and cmMTB-treated macrophages at day 3. **(B)** Gene set enrichment plot of the glycolysis genes (hallmark collection of MSigDB). This plot shows the distribution of the glycolysis gene set between macrophages exposed to cmCTR versus cmMTB for 3 d. The skewing of the genes to the right and the negative normalized enrichment score (NES = –1.66) indicate enrichment of genes related to glycolysis in macrophages exposed to cmMTB versus cmCTR (FDR = 0.005). **(C)** Glucose uptake was assessed by flow cytometry using 2-NBDG (2-(7-nitro-2,1,3-benzoxadiazol-4-yl)-D-glucosamine) staining in cells treated with cmCTR versus cmMTB for 3 d. Representative histograms (left) and dot plots showing the geomean fluorescence intensity (MFI). **(D, E)** Supernatant of cmCTR- and cmMTB-treated cells was collected at different time points posttreatment, and glucose (D) and lactate (E) flux (mmol.(L.h)$^{-1}$) was assessed by RMN analysis for n = four donors. **(F, G)** Dot plots showing the concentration (mM) of lactate in the supernatants at day 3. **(H)** HIF-1α expression was assessed by flow cytometry at day 3. The mean ± SD is shown. **(C, D, E, F, G, H)** Each circle represents a single donor. Histograms represent mean values. **(I, J)** HIF-1α localization was assessed by IF in macrophages exposed to cmMTB versus cmCTR. **(I)** Representative immunofluorescence images: HIF-1α (green) and nuclei (DAPI, blue). Scale bar, 10 μm. White arrowheads show HIF-1α translocation in the nucleus. **(J)** Quantification of the ratio of HIF-1α signal intensity in the nucleus versus the cytoplasm. 20 cells/conditions, n = 3 donors. Statistical analysis: t test data with normal distribution; *P ≤ 0.05; **P ≤ 0.01; ***P ≤ 0.001; n.s., not significant.

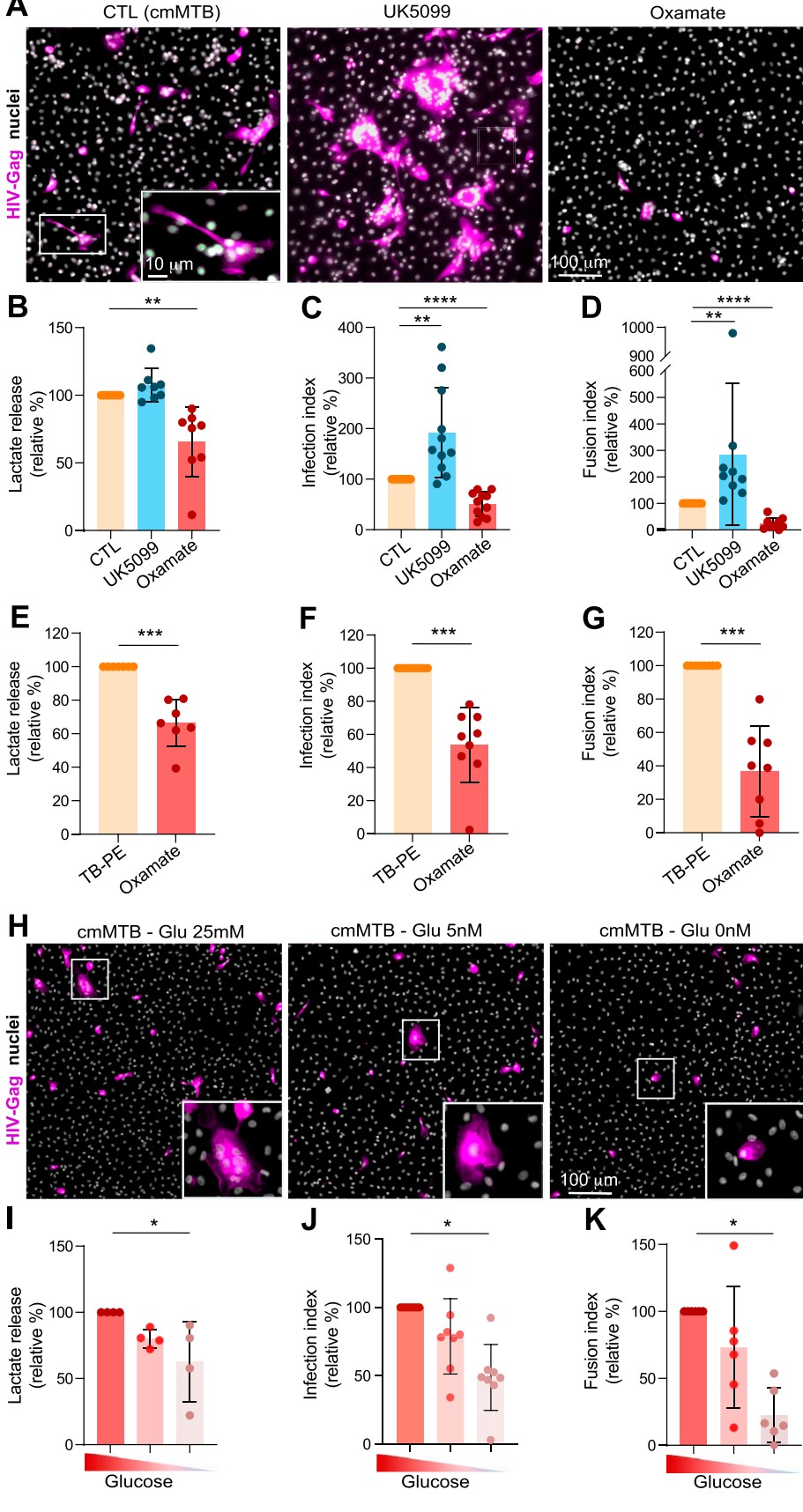

**Figure 3. Modulation of aerobic glycolysis impacts HIV-1 infection of macrophages in TB-derived environments.**

Monocytes from healthy subjects were differentiated in conditioned medium from Mtb-infected macrophages (cmMTB) or with pleural effusions (PE) from TB (TB-PE) for 3 d. At day 2 of differentiation, metabolic inhibitors were added to the culture medium. At day 3, cells were infected by HIV-1 (NLAD8-VSVG) (for three further days), and HIV-1 infection of monocyte-derived macrophages (MDM) was measured (see the experimental design, Fig S2A and B). **(A, B, C, D)** Analysis of MDM infection upon UK5099 or oxamate treatment. **(A)** Representative IF images: HIV-Gag (magenta) and nuclei (DAPI, gray). Scale bar, 100 μm. **(B)** Lactate release measured at day 3 (24 h after drug treatment); quantification of MDM infection index (C, n = 9–11 donors) and MDM fusion index ((D), n = nine donors), normalized to the control condition (CTL = cmMTB w/o treatment). **(E, F, G)** Analysis of HIV-1 infection of TB-PE–treated macrophages upon oxamate treatment by microscopy. **(E)** Lactate release measured at day 3 (24 h upon treatment); quantification of MDM infection index ((F), n = nine donors) and MDM fusion index ((G), n = eight donors), normalized to the control condition (CTL = cmMTB without treatment). **(H, I, J, K)** Analysis of MDM infection upon glucose deprivation (normal condition: 25, 5, or 0 mM glucose) at day 2. **(H)** Representative IF images: HIV-Gag (magenta) and nuclei (DAPI, gray). Scale bar, 100 μm. **(I)** Lactate release measured at day 3 (24 h upon treatment); quantification of MDM infection index ((J), n = six donors) and MDM fusion index ((K), n = six donors), normalized to the control condition (CTL = cmMTB without treatment). Mean ± SD is shown. Each circle represents a single donor. Statistical analysis: data with normal distribution; *$P$ ≤ 0.05; **$P$ ≤ 0.01; ***$P$ ≤ 0.001; ****$P$ ≤ 0.0001.

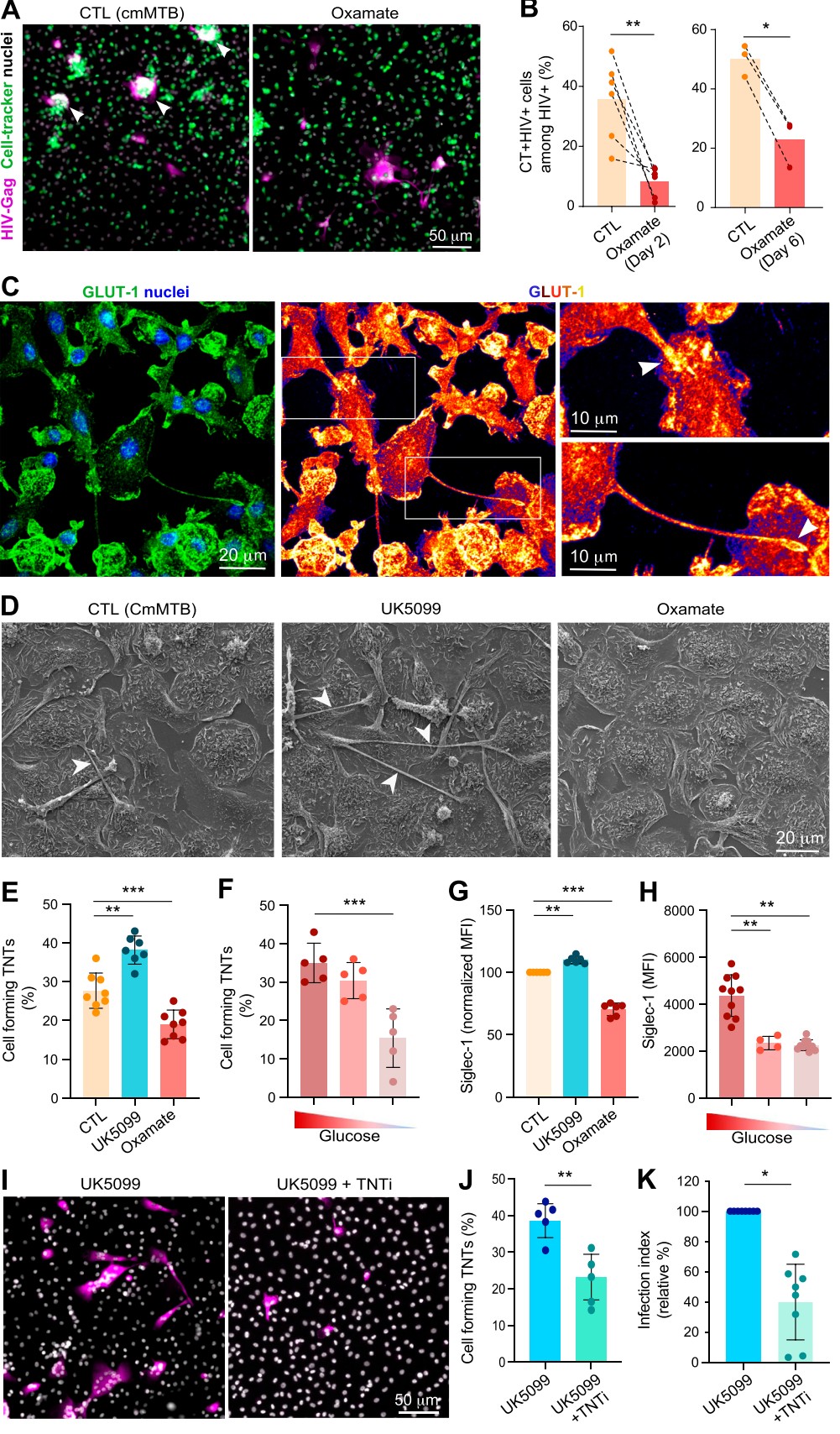

**Figure 4. Glycolysis potentiates HIV-1 spread between macrophages in TB-derived environments.**
**(A, B)** Glycolysis favors HIV-1 transfer between macrophages.
**(A)** Representative IF images of cocultures (see the experimental design, Fig S3A). HIV-1 Gag (magenta), CellTracker (green), and DAPI (gray). Scale bar, 50 μm. Arrowheads show multinucleated double-positive cells. **(B)** Quantification of the percentage of CellTracker+ cells among Gag+ cells in the condition of oxamate treatment at day 2 (left, n = 6 donors) or at day 6 (right, n = 3 donors). Histograms represent mean values. **(C)** Confocal images of HIF-1α localization in cmMTB-treated cells. Left panel: HIF-1α (green) and nuclei (DAPI, blue); see also Video 2. Scale bar, 20 μm. Right panels: intensity profiles of HIF-1α staining (Fire LUT). The white arrowhead shows the accumulation of HIF-1α at the tip of tunneling nanotube (TNT). Scale bars, 10 μm. **(D, E, F, G, H)** Glycolysis regulates HIV-1–induced TNT formation and Siglec-1 surface expression, impacting HIV-1 dissemination in macrophages. CmMTB-treated cells were infected with HIV-1 at day 3, and TNT formation and Siglec-1 expression in MDMs were assessed at day 6. **(D, E, F)** Analysis of TNT formation upon UK5099 or oxamate treatment in comparison with the control condition (CTL = cmMTB without treatment). **(D)** Representative scanning electron microscopy images. Scale bar, 20 μm. **(E)** Quantification of the percentage of cells forming TNTs (n = eight donors). **(F)** Analysis of TNT formation after glucose deprivation (5 or 0 mM glucose) for 2 d in comparison with the control condition (CTL = cmMTB without treatment; normal condition 25 mM glucose). Quantification of the percentage of cells forming TNTs (n = five donors, at least 200 cells/condition/donor). **(G, H)** Analysis of Siglec-1 expression at the surface of MDMs after UK5099/oxamate treatment (G) and glucose deprivation (H). Geomean fluorescence intensity (MFI) of Siglec-1 cell-surface expression, normalized to the control condition (CTL = cmMTB without treatment) in (D). **(I, J, K)** Analysis of HIV-1 infection of MDMs. CmMTB-treated cells were exposed to UK5099 and infected with HIV-1 in the presence or absence of TNT inhibitor (TNTi, 20 μM). **(I)** Representative IF images: HIV-Gag (magenta) and nuclei (DAPI, gray). Scale bar, 50 μm. Quantification of TNT formation ((J), n = five donors) and infection fusion index ((K), n = eight donors), normalized to the control condition (cmMTB-treated cells in the presence of UK5099). Means ± SD are shown. Each circle represents a single donor. Statistical analysis: data with normal distribution; *P ≤ 0.05; **P ≤ 0.01; ***P ≤ 0.001.

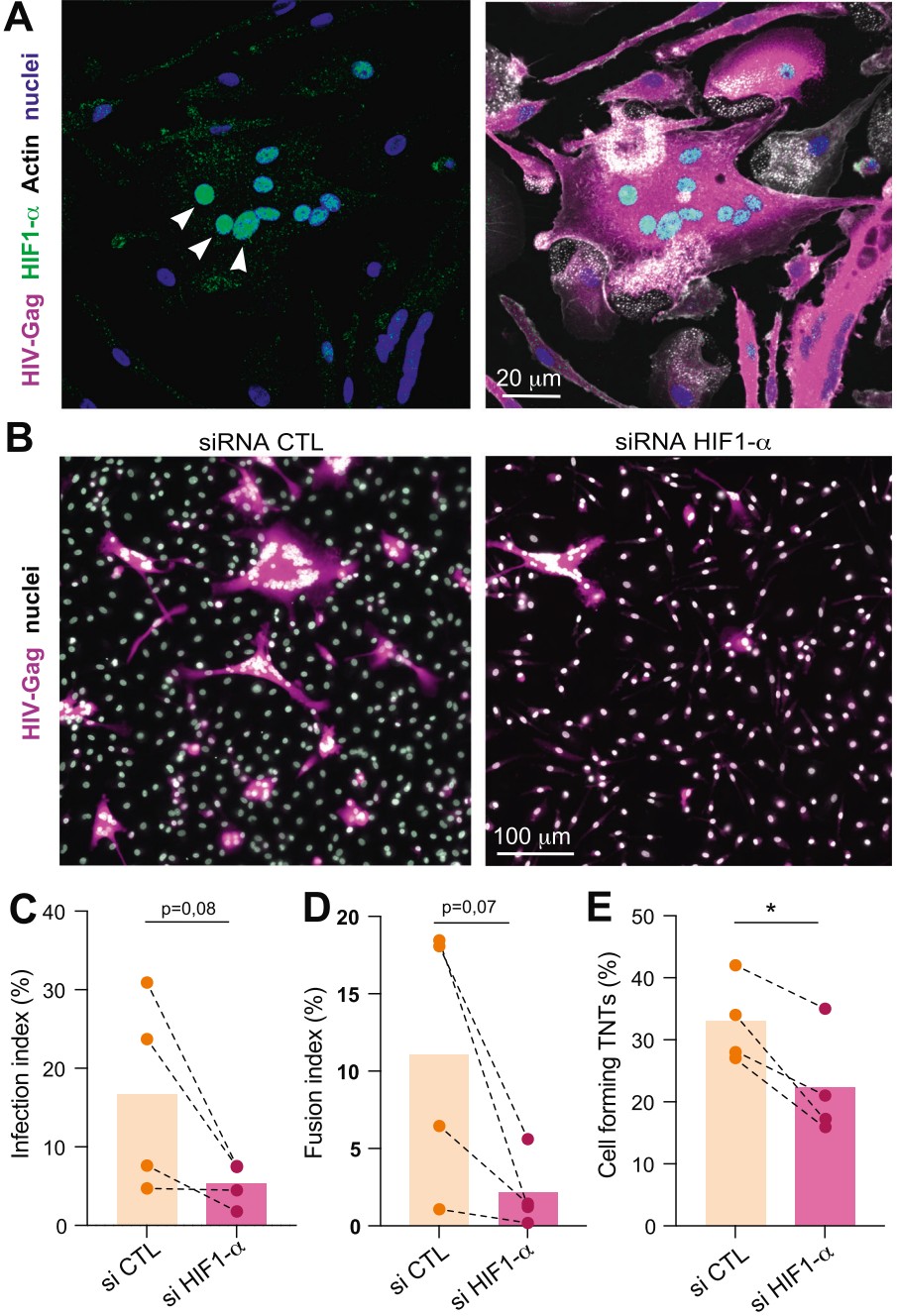

**Figure 5. HIF-1α regulates tunneling nanotube (TNT) formation and HIV-1 dissemination in macrophages in TB-derived environments.**
**(A)** HIF-1α is activated in HIV-positive macrophages. Human monocytes were differentiated with cmMTB for 3 d and infected with HIV-1 NLAD8-VSVG for an additional 3 d. Representative confocal image of HIV-positive multinucleated giant cell (MGC) with HIF-1α translocation in the nuclei: HIF-1α (green), HIV-Gag (magenta), actin (gray), and nuclei (DAPI, blue). Scale bar, 20 μm. **(B, C, D, E)** Analysis of HIV-1 infection and TNT formation upon siRNA against HIF-1α by microscopy. Briefly, monocytes from healthy subjects were transfected with siRNA against HIF-1α (or control siRNA), incubated with cmMTB for 3 d, and then infected with HIV-1 NLAD8-VSVG for an additional 3 d. **(B)** Representative IF images: HIV-Gag (magenta) and nuclei (DAPI, gray). Scale bar, 100 μm. **(C, D, E)** Quantification of the infection index ((C), n = 4 donors), fusion index ((D), n = 4 donors), and TNT formation ((E), n = 4 donors). Statistical analysis: data with normal distribution; *$P$ ≤ 0.05; **$P$ ≤ 0.01; ***$P$ ≤ 0.001; ****$P$ ≤ 0.0001.

These macrophages are notably abundant in coinfection sites, such as the pulmonary pleural cavity in patients and the lung parenchyma in NHPs (Lastrucci et al, 2015; Souriant et al, 2019; Dupont et al, 2020, 2022). The present study sheds light on how TB exacerbates HIV-1 infection in our immunomodulatory macrophage model, emphasizing the critical role of macrophage immunometabolism in the context of HIV-1/Mtb coinfection and the essential role of glycolysis in TNT formation. Understanding this mechanism is crucial, as it highlights potential dangers for HIV patients receiving the TB vaccine and underscores the need to identify new targets for vaccines or therapies

aimed at preventing TB-induced deleterious immunomodulatory macrophages.

Mtb is known to alter macrophage metabolism, but findings across studies vary because of differences in Mtb strains, timing of metabolic analysis, and macrophage origins (Cumming et al, 2018; Kumar et al, 2019; Llibre et al, 2021; Olson et al, 2021). In general, it is well accepted that infected macrophages undergo aerobic glycolysis as part of the Warburg effect to eliminate Mtb and other intracellular pathogens (Lugo-Villarino & Neyrolles, 2014). This shift toward glycolysis is evident in studies showing GlycoATP

production boosts in macrophages challenged with noninfectious Mtb strains (Gleeson et al, 2016; Marin Franco et al, 2020). Many studies report increased glycolysis postinfection with virulent Mtb strains, but often do not distinguish between infected and bystander cells (Braverman et al, 2016; Huang et al, 2019; Russell et al, 2019; Shi et al, 2019; Marin Franco et al, 2020). The use of TB-associated microenvironments enables the study of the bystander effect induced by infected cells on the metabolic state of their neighbor cells, including recruited circulating monocytes at the infection site. Using in vitro (cmMTB) and ex vivo (PE-TB) models to mimic TB-associated microenvironments, we observed increased glycolysis in monocytes differentiating into immunomodulatory macrophages, as measured by several metabolic analyses, an enriched glycolytic gene signature, and activation of the metabolic regulator HIF-1α. Although treating monocytes with CmMTB or TB-PE leads to an immunomodulatory macrophage phenotype dependent on the IL-10/STAT3 (Lastrucci et al, 2015; Souriant et al, 2019) and type I interferon (IFN-I)/STAT1 (Dupont et al, 2020, 2022) signaling pathways, the literature is well documented that IL-10 and IFN-I are associated with the suppression of glycolytic activity (Yeh et al, 2018; Olson et al, 2021). This evidence suggests that these cytokines alone are unlikely to be responsible for the metabolic changes observed in macrophages. We infer that mycobacterial antigens in the fluids might drive glycolysis in cells, as attenuated Mtb strains consistently induce glycolysis in macrophages, unlike live or virulent strains. Although this work did not focus on the impact of the glycolytic shift on Mtb control, it may contribute to high intracellular replication of the bacillus reported in these permissive macrophages (Lastrucci et al, 2015). This supports the notion that increased glycolysis does not necessarily improve infection control. Recent studies have shown that monocytes from TB patients, inherently biased toward glycolysis, differentiate into dendritic cells that do not effectively engage glycolytic flux, leading to reduced migratory capacity (Maio et al, 2024). Thus, despite this shift toward increased glycolysis, prior exposure to inflammatory signals may render macrophages unable to undergo the strong and lasting glycolytic reprogramming observed in inflammatory M1 macrophages that control intracellular Mtb replication. Future investigations shall address this important issue and establish in vivo correlations between the glycolytic shift and the macrophage compartment in biopsies and samples from NHP and TB patients. However, modulation of glycolytic activity in alveolar and interstitial macrophages in Mtb-infected mice has already been shown to be important in controlling bacterial growth (Huang et al, 2019).

Viruses lack the metabolic machinery for survival and have evolved strategies to exploit their host cells' metabolic resources, as reported for HIV-1 (Saez-Cirion & Sereti, 2021). Our understanding of these metabolic adaptations primarily comes from studies of CD4⁺ T lymphocytes, the most extensively studied host cells for this virus. HIV-1 fitness is favored in T lymphocytes with high metabolic activity (Shehata et al, 2017; Valle-Casuso et al, 2019). The metabolic program of different T-cell subtypes is crucial in determining their susceptibility to infection, regardless of their activation level, with glycolysis and glutaminolysis being key to sustaining the pre-integration steps of HIV-1 infection (Loisel-Meyer et al, 2012; Palmer et al, 2016; Shehata et al, 2017; Clerc

et al, 2019; Valle-Casuso et al, 2019). Under specific conditions in TB-associated microenvironments, we observed that the early steps of the viral cycle are not affected by glycolysis inhibition in macrophages. However, the metabolic activity of CD4⁺ T cells is critical not only for facilitating intracellular replication during HIV-1 infection but also for governing the overall infection process (Shehata et al, 2017). For myeloid cells, previous studies have shown that CD16⁺ monocytes, the most permissive monocyte subset to HIV-1 infection (Ellery et al, 2007; Rodrigues et al, 2017), exhibit heightened expression of Glut-1, increased glucose uptake, and elevated lactate release (Palmer et al, 2016). However, a direct link between glycolysis and monocyte/macrophage susceptibility to HIV-1 was lacking.

In this study, using drugs to selectively modulate the glycolytic pathway, we demonstrated that this pathway is involved in HIV-1 infection and the formation of MGCs, hallmarks of HIV-1 infection of macrophages (Verollet et al, 2015; Han et al, 2022). In TB-associated microenvironments, oxamate, a competitive inhibitor of lactate dehydrogenase A (LDHA), reduced the number of infected MGCs by more than twofold. Oxamate also blocks lactate production without altering pyruvate's role as a fuel for mitochondrial respiration, distinguishing the contribution of glucose to ATP production by glycolysis versus OXPHOS. Although the use of 2-DG has been controversial (Saez-Cirion & Sereti, 2021), this widely used glycolysis inhibitor yielded the same results. Complementing pharmacological inhibitor studies, glucose deprivation experiments confirmed our results. In particular, we could not restore HIV-1 infection when galactose was used in place of glucose to generate ATP only by OXPHOS. Similar findings have been reported in previous studies, highlighting the importance of macrophage glycolytic activity in viral replication, as observed for dengue (Fontaine et al, 2015) and murine norovirus (Passalacqua et al, 2019). Interestingly, during vesicular stomatitis virus (VSV) infection, glycolysis promotes viral replication by negatively regulating IFN-I and antiviral responses (Zhang et al, 2019). Because immunoregulatory macrophages exhibit a strong but defective IFN-I signature (Dupont et al, 2022), it is plausible that glycolysis may impair the IFN-I antiviral response, and vice versa (Maio et al, 2024). To explore whether environmental triggers of glycolysis influence HIV-1 infection in macrophages and broader HIV-1 pathogenesis, additional research is needed beyond the context of Mtb coinfection. Current knowledge indicates that HIV-1 infection can enhance glucose metabolism in macrophages (Datta et al, 2016). However, this effect may vary depending on the macrophage type, state, and infection timing (Hollenbaugh et al, 2011; Castellano et al, 2019). Although we did not examine the metabolic status of macrophages post-HIV-1 infection alone, sustained glycolytic activity might optimize long-term HIV-1 infection. Another aspect to consider is how macrophage polarization impacts susceptibility to HIV-1 (Cassol et al, 2009, 2010; Hendricks et al, 2021). Interestingly, polarized macrophages, which have distinct metabolic profiles, are less susceptible to HIV-1 infection than unpolarized macrophages (Saha et al, 2017). However, these studies primarily used cell-free viral infection, which is less common in vivo than cell-to-cell infection (Bracq et al, 2018; Han et al, 2022; Mascarau et al, 2023). Future research should focus on the link between metabolism and heterotypic intercellular HIV-1 infection, especially the transfer

from T cells to macrophages, which is influenced by macrophage polarization (Mascarau et al, 2023).

As mentioned above, cell-to-cell transfer of HIV-1 is more efficient than infection with cell-free virus and plays a critical role in virus dissemination in vivo, especially in macrophages (Murooka et al, 2012; Sewald et al, 2012; Dupont & Sattentau, 2020; Han et al, 2022). The primary mechanism for this transfer involves TNTs, which facilitate the transport of viral particles. TNT formation is exacerbated in TB-associated microenvironments enriched with IL-10. However, the molecular mechanisms behind TNT formation are not fully understood (Eugenin et al, 2009; Hashimoto et al, 2016; Okafo et al, 2017; Souriant et al, 2019; Uhl et al, 2019; Dupont et al, 2020; Lotfi et al, 2020). We showed that TNT formation between infected macrophages not only contributes to their fusion into MGCs but also that glycolysis plays a crucial role in controlling their formation. Blocking glycolysis reduces TNT formation, whereas promoting glycolysis increases its presence. TNTs are F-actin–based, open-ended, membranous channels that connect cells over varying distances. Their formation and stability can be influenced by extracellular conditions, such as nutritional deprivation, oxidants, acidic conditions, and cytokines (Lou et al, 2012; Goodman et al, 2019). In line with this, we now identify GLUT-1 enrichment at the tips of macrophage TNTs, suggesting localized glucose uptake during TNT formation. Based on prior work showing that actin-based protrusions require local ATP production for extension and contractile dynamics, glucose entry at TNT tips may directly support the energetic demands of TNT growth and maintenance. Collectively, our work highlights the role of intracellular metabolism, particularly glycolysis, in promoting TNT formation in macrophages. However, whether these findings can be translated to other cell types that form TNTs remains an open question, as studies in cancer cells and mesenchymal stem cells have shown contradictory results in TNT formation and dynamics (Liu et al, 2014; Thayanithy et al, 2014).

In macrophages, two types of TNTs exist: thin TNTs (<0.7 $\mu$m in diameter, containing F-actin) and thick TNTs (>0.7 $\mu$m in diameter, rich in F-actin and microtubules) (Onfelt et al, 2006; Souriant et al, 2019). Glycolysis modulation affects both types, with increased glycolytic activity in immunoregulatory macrophages potentially providing the energy needed for actin cytoskeletal rearrangements essential for TNT formation. ATP is crucial for supporting cellular functions involving actin remodeling, such as cell migration and epithelial-to-mesenchymal transition (DeWane et al, 2021). In HIV-1–infected macrophages, ATP is also vital for the release of particles from virus-containing compartments (Graziano et al, 2015). Our study suggests that glycolysis, by promoting TNT formation, enhances HIV-1 dissemination between macrophages. Besides HIV-1, thick TNTs can transfer various organelles, including mitochondria, which can alter the metabolism and functional properties of recipient cells (Hekmatshoar et al, 2018; Wang et al, 2021; Goliwas et al, 2023), further complicating the model. Within the context of thick TNTs, Siglec-1 (a sialic acid–binding lectin) plays a critical role in their stabilization (Dupont et al, 2020). We show that the expression of Siglec-1 is influenced by glycolytic activity within macrophages. High levels of glycolysis up-regulate Siglec-1 expression, thereby enhancing the formation and stability of thick TNTs and promoting efficient viral dissemination. Given that

*SIGLEC-1* is an IFN-stimulated gene (Dupont et al, 2022), it is plausible that glycolysis influences the capacity of bystander macrophages to produce autocrine IFN$\beta$ (Olson et al, 2021). This metabolic regulation underscores the significance of glycolysis in modulating macrophage functions and intercellular communications, particularly in the context of viral infections. A deeper understanding of the mechanisms of cell-to-cell communication mediated by TNTs and of the influence of metabolism on these processes is still needed.

Beyond the metabolic control of TNT formation, our findings contribute to TNT biology by identifying a transcriptional link between metabolic rewiring and the structural machinery that supports intercellular connectivity. Although TNT formation has classically been associated with cytoskeletal regulators and membrane remodeling factors (Dupont et al, 2018; Rey-Barroso et al, 2024), there has been no single transcriptional program clearly tied to TNT biogenesis across contexts. HIF-1$\alpha$ is a master regulator of glycolytic metabolism and cellular adaptation to stress, known to drive expression of glycolytic enzymes and influence macrophage energetics and motility (Cramer et al, 2003; Qiu et al, 2023). Although we have not directly demonstrated that HIF-1$\alpha$ binds promoters of specific TNT effectors, the association between HIF-1$\alpha$ nuclear localization in fused macrophages and the dependence of TNT formation on glycolytic activity suggests that HIF-1$\alpha$ may influence TNT biogenesis via its control of metabolic programs rather than through purely pro-inflammatory signaling. Indeed, glycolysis has been shown in other systems to support membrane protrusions and energy-intensive cytoskeletal remodeling (Ozawa et al, 2015; Semba et al, 2016), processes that are critical for both TNT extension and cell fusion. This places HIF-1$\alpha$ conceptually upstream of metabolic pathways that create permissive conditions for intercellular structures such as TNTs, expanding the field's understanding of how metabolic transcriptional regulation intersects with long-range cell–cell communication.

To conclude, our study demonstrates that TB-associated microenvironments induce aerobic glycolysis in immunoregulatory macrophages, increasing their propensity to form TNTs and facilitating HIV-1 transfer between macrophages. By identifying HIF-1$\alpha$ as a transcriptional regulator associated with this metabolic state, our work adds a mechanistic layer linking glycolytic rewiring to the cellular processes that support TNT biogenesis and macrophage fusion, key features of cell-to-cell viral dissemination. Although a direct role of HIF-1$\alpha$ in TNT formation remains to be formally established, its involvement is most consistent with a metabolic rather than a pro-inflammatory control of these energy-demanding processes. The role of glycolysis during viral infection remains unclear, especially regarding its potential dysregulation in coinfection with other pathogens. Recent reports indicate that glycolysis in tissue macrophages, which exhibit an intermediate M1/M2 profile, is necessary for HIV-1 reactivation from latency (Real et al, 2022). This suggests glycolysis could be a potential target in future HIV-1 eradication strategies. Our findings support the idea that targeting glycolysis could disrupt viral progression during Mtb coinfection. Further research is necessary to fully understand the metabolic effects of TB-associated microenvironments on the

lung macrophage compartment. We speculate that identifying the mycobacterial antigen in the TB-associated milieu driving glycolysis would enable an extremely targeted therapy to specifically inhibit this metabolic rewiring in human macrophages. This will provide opportunities to disentangle the complex beneficial and detrimental roles that the glycolysis/OXPHOS balance plays in immunity and coinfection with HIV-1.

# Materials and Methods

### Human subjects

Human primary monocytes were isolated from healthy subject (HS) buffy coat (provided by Etablissement Français du Sang, Toulouse, France, under contract 21/PLER/TOU/IPBS01/20,130,042) and differentiated toward macrophages (Souriant et al, 2019). According to articles L12434 and R124361 of the French Public Health Code, the contract was approved by the French Ministry of Science and Technology (agreement number AC 2009921). Written informed consent forms were obtained from the donors before sample collection.

PE samples from patients with TB were obtained by physicians at Hospital F. J. Muñiz via therapeutic thoracentesis. The diagnosis of TB pleurisy was based on a positive Ziehl–Neelsen stain or Lowenstein–Jensen culture from pleural fluid and/or histopathology of a pleural biopsy, and was further confirmed by an Mtb-induced IFN-γ response and an adenosine deaminase–positive test. Exclusion criteria included a positive HIV test and the presence of concurrent infectious diseases or noninfectious conditions (cancer, diabetes, or steroid therapy). None of the patients had multi-drug-resistant TB. The research was carried out in accordance with the Declaration of Helsinki (2013) of the World Medical Association and was approved by the Ethics Committee of the Hospital F. J Muñiz (protocol number: NIN-2601-19). Written informed consent was obtained before sample collection.

### Bacteria

Mtb (H37Rv; see Table 1) was grown at 37°C in Middlebrook 7H9 medium, supplemented with 10% albumin–dextrose–catalase, as described previously (Lastrucci et al, 2015). Exponentially growing Mtb was centrifuged ($460g$) and resuspended in PBS ($MgCl_2$, $CaCl_2$-free; Gibco). Clumps were dissociated by 20 passages through a 26-G needle and then resuspended in RPMI 1640 containing 10% FBS. Bacterial concentration was determined by measuring the optical density (OD) at 600 nm.

### Viruses

Viral stocks were generated by transient transfection of 293T cells with the proviral plasmids encoding HIV-1 NLAD8, kindly provided by Serge Bénichou (Institut Cochin, Paris, France), with the VSVG envelope, as previously described (Mascarau et al, 2023) (Table 1). Supernatants were harvested 48 h post-transfection, and p24 antigen concentration was assessed by a homemade ELISA. HIV-1 infectious units were quantified using TZM-bl cells as previously reported (Mascarau et al, 2023).

### Human monocyte–derived macrophage culture

Monocytes were isolated and differentiated toward monocyte-derived macrophages as described previously (Lastrucci et al, 2015; Souriant et al, 2019). Briefly, peripheral blood mononuclear cells were recovered by gradient centrifugation on Ficoll-Paque Plus (GE Healthcare). $CD14^+$ monocytes were then isolated by positive selection magnetic sorting, using human CD14 Microbeads and LS columns (Miltenyi Biotec). Cells were then plated at $1.6 × 10^6$ cells in six-well plates and allowed to differentiate for 5–7 d in RPMI-1640 medium (Gibco), 10% FBS (Sigma-Aldrich), and human M-CSF (20 ng/ml; PeproTech) before infection with Mtb H37Rv for conditioned-medium preparation. The cell medium was renewed every 3rd or 4th d.

### Preparation of conditioned media of Mtb-infected macrophages

CmMTB was prepared as reported previously (Lastrucci et al, 2015). Briefly, MDMs were infected with Mtb H37Rv at an MOI of 3. After 18 h of infection at 37°C, culture supernatants were collected and filtered by double filtration (0.2-μm pores), and aliquots were stored at –80°C. CmCTR was obtained from uninfected macrophages.

### Treatment of monocytes with the secretome of Mtb-infected macrophages or pleural effusion from TB patients

Freshly isolated $CD14^+$ monocytes from HS were allowed to adhere in the absence of serum ($4 × 10^5$ cells in 50 μl in 24-well plates or $2 × 10^6$ cells in 1.5 ml in six-well plates). After 1 h, cmMTB or cmCTR supplemented with 20 ng/ml|l M-CSF and 20% FBS were added to the cells (vol/vol). For experiments with PE, samples were collected in heparin tubes and centrifuged at $300g$ for 10 min at RT. The cell-free supernatants were transferred into new plastic tubes and further centrifuged at $12,000g$ for 10 min, and aliquots were stored at –80°C. After the PE samples were diagnosed, pools were prepared by mixing equal amounts of the individual PE associated with a specific etiology. The pools were decomplemented at 56°C for 30 min and filtered through 0.22-μm pores to remove any remaining debris or residual bacteria. A pool of PE samples from 10 patients with active TB was prepared. In addition, a pool of PE from five patients with transudative heart failure was included as a control. Both pools were supplemented with 40 ng/ml|l M-CSF, and 40% FBS was added to the cells (25% vol/vol). Cells were then cultured for 3 d. Cell-surface expression of macrophage activation markers was measured by flow cytometry using standard procedures.

### HIV-1 infection

At day 3 of differentiation, $0.5 × 10^6$ macrophages in 24-well plates were infected with HIV-1 VSVG (or HIV-Gag-iGFP-VSVG for

**Table 1. Key resources and reagents used in this study.**

| Reagent type (species) or resource | Source or reference | Identifiers |
|---|---|---|
| **Critical commercial assays** | | |
| Lactate assay kits | Wiener | Cat# 1999795 |
| 2-NBDG | Invitrogen | Cat# N13195 |
| MitoSOX Red Mitochondrial Superoxide Indicator | Invitrogen | Cat# M36008 |
| CellTracker Green | Thermo Fisher Scientific | Cat# C7025 |
| MitoTracker Deep Red | Thermo Fisher Scientific | Cat# M22426 |
| CellROX Deep Red | Thermo Fisher Scientific | Cat# C10422 |
| Mouse anti-human CD14 Microbeads | Miltenyi Biotec | Cat# 130-050-201 |
| LS magnetic columns | Miltenyi Biotec | Cat# 130-042-401 |
| Cell dissociation buffer | Thermo Fisher Scientific | Cat# 13151014 |
| Phalloidin Alexa Fluor 488 | Thermo Fisher Scientific | Cat# A12379 |
| CellTracker Green CMFDA Dye | Thermo Fisher Scientific | Cat# C7025 |
| Fluorescence Mounting Medium | Agilent Technologies | Cat# S302380-2 |
| Ficoll-Paque Plus | Cytiva | Cat# 17144003 |
| Trypsin–EDTA (0.05%) | Gibco | Cat# 25300054 |
| RPMI 1640 medium, no glucose | Thermo Fisher Scientific | Cat# 11-879-020 |
| Glucose solution | Thermo Fisher Scientific | Cat# A2494001 |
| **Bacterial and virus strains** | | |
| HIV-1 NLAD8-VSVG | Gift from Dr. S Benichou, Institut Cochin, Paris, France | N/A |
| HIV-1 ADA Gag-iGFP-VSVG | Gift from Dr. P. Benaroch, Institut Pasteur, Paris, France | N/A |
| *M. tuberculosis* H37Rv | N/A | N/A |
| **Biological samples** | | |
| Buffy coat from healthy donors | Etablissement Français du Sang, Toulouse, France | N/A |
| Patient-derived pleural effusions | Hospital F. J Muñiz (Buenos Aires, Argentina) | N/A |
| **Antibodies** | | |
| Anti-human HIF-1α | Cell Signaling | Cat# 36169S |
| LIVE/DEAD Fixable Aqua Dead Cell | Thermo Fisher Scientific | Cat# L34957 |
| Mouse monoclonal anti-HIV-1 p24 (clone KC57, FITC- or RD1-coupled) | Beckman Coulter | Cat# 6604665/7 |
| Anti-human CD16 | BioLegend | Clone 3G8 |
| Anti-human CD163 | BioLegend | Clone GHI/61 |
| Goat anti-mouse IgG, Alexa Fluor 488 | Thermo Fisher Scientific | Cat# A-10684 |
| Goat anti-mouse IgG, Alexa Fluor 555 | Cell Signaling Technology | Cat# 4409 |
| Anti-human CD169 | BioLegend | Clone 7-239 |

**Table 1. Continued**

| Reagent type (species) or resource | Source or reference | Identifiers |
|---|---|---|
| **Critical commercial assays** | | |
| Anti-human GLUT-1 | Abcam | ab115730 |
| Chemicals, peptides, and recombinant proteins | | |
| Human M-CSF | PeproTech | Cat# 300-25 |
| TNTi | Pharmeks | N/A |
| Roswell Park Memorial Institute (RPMI 1640) medium | Gibco | Cat# 21875034 |
| Triton X-100 | Sigma-Aldrich | Cat# T8532 |
| IF (0.3%) | | |
| WB (1%) | | |
| Tween-20 | Sigma-Aldrich | Cat# P9416 |
| UK5099 | MedChemExpress | Cat# HY-15475 |
| GSK2837808A | MedChemExpress | Cat# HY-100681 |
| Oxamate | Santa Cruz Biotechnology | Cat# sc-215880A |
| 2-DG | MedChemExpress | Cat# HY-13966 |
| FBS | Sigma-Aldrich | N/A |
| Ficoll-Paque Plus | Cytiva | Cat# 17144003 |
| PBS | Gibco | Cat# 14190144 |
| BSA | Euromedex | Cat# 04-100-812-C |
| Tris-buffered saline | Euromedex | Cat# 2-9134-10 |
| PFA | Delta-Microscopies | Cat# D15714 |
| Sucrose | Sigma-Aldrich | Cat# S0389 |
| Glutaraldehyde | Delta-Microscopies | Cat# D16220 |
| Galactose | Sigma-Aldrich | Cat# G0750 |
| Software and algorithms | | |
| ImageJ | ImageJ | www.imagej.nih.gov/ij |
| Prism (v9) | GraphPad | www.graphpad.com |
| FACSDiva | BD Bioscience | http://www.bdbiosciences.com/ |
| FlowJo v10 | FlowJo | https://www.flowjo.com/ |
| ZEN Black | Zeiss | https://www.zeiss.fr/microscopie/produits/microscope-software/zen.html |
| siRNA | | |
| HIF-1α human | Dharmacon | L-004018-00-0010 |

live imaging) strain at an MOI of 1 in a fresh culture medium. HIV-1 infection and replication were assessed at day 3 postinfection by measuring p24-positive cells by immunostaining or flow cytometry, as described previously (Mascarau et al, 2020).

## Coculture assays

Half of the cmMTB-treated macrophages were infected with HIV-1 NLAD8 VSVG for 3 d. At day 6, the other half of the cell population was stained with CellTracker Green CMFDA Dye (Thermo Fisher Scientific). Thereafter, cells were washed three times with PBS, Mg2+/Ca2+, and detached using trypsin 0.25%/EDTA (Gibco) for 15 min. Then, they were cocultured at a 1:1 ratio on glass coverslips in a 24-well plate for 24 h. They were fixed with PFA 3.7% and sucrose 15 mM in PBS for 1 h, and HIV-1 transfer was assessed by immunofluorescence as described previously (Souriant et al, 2019). In defined experimental kinetic conditions, oxamate was added 24 h before HIV-1 infection (day 2) or 30 min after coculture (day 6).

**Drug treatments and glucose deprivation experiments**

Two days after purification, monocytes were treated with 20 mM sodium oxamate (Santa Cruz Biotechnology), 50 $\mu$M UK5099 (MedChemExpress), 60 $\mu$M GSK 2837808A (MedChemExpress), or 500 $\mu$M 2-DG (MedChemExpress). For the experiments using TNTi (Pharmeks), cells were treated with 20 $\mu$M TNTi at days 0 and 3, as described previously (Souriant et al, 2019). For glucose deprivation experiments, the medium was changed at day 2 of differentiation to RPMI 1640 medium without glucose (Thermo Fisher Scientific) and FBS, supplemented with 0.5 or 25 mM glucose (Thermo Fisher Scientific) or 25 mM galactose (Sigma-Aldrich).

**RNA interference**

CD14[+] human monocytes were transfected with 200 nM siRNA against HIF-1$\alpha$ using the HiPerfect system (QIAGEN) as described previously (Lastrucci et al, 2015). The mix of HiPerfect and siRNA was incubated for 15 min at RT, and then, the cells were added drop by drop. The following siRNAs (Dharmacon) were used: human ON-TARGET plus SMART pool siRNA nontargeting control pool (siCTL); human ON-TARGET plus SMART pool siRNA targeting HIF-1$\alpha$ sequences: 5′-CGU AUGCUGUCCAGUCUAA-3′; 5′-GAGGGAAGUUUGGUUCUUU-3′; 5′-UCG CAAGCCUGAUACCAUU-3′; 5′-GGCUGAAACUCAAUAAGAA-3′.

**Immunofluorescence microscopy**

Cells were fixed with 3.7% PFA and 15 mM sucrose in PBS. Cells were permeabilized with 0.3% Triton X-100 for 10 min and saturated with 1% BSA in PBS for 30 min (see Table 1 for antibodies and reagents). Cells were incubated with anti-HIV-1-Gag KC57 antibody RD1 (1:100) in PBS/BSA 1% for one h, washed, and then incubated with Alexa Fluor 555 goat anti-mouse IgG secondary antibody (1:1,000), Alexa Fluor 488 Phalloidin (1:500), or WGA 488 and DAPI (500 ng/ml) in PBS/BSA 1% for 30 min. For microtubule staining, cells are incubated with anti-tubulin antibodies (1:100, Sigma-Aldrich). In addition, anti-HIF-1$\alpha$ antibody (clone D1S7W, 1:100) and anti-GLUT1 antibody (1:100) were used (see Table 1). Coverslips were mounted on a glass slide using Fluorescence Mounting Medium (Dako). Images for quantification of infection were acquired using a Zeiss Axio Imager M2 and a 20×/0.8 Plan Apochromat or 40×/0.95 Plan Apochromat objectives (Zeiss). Images were acquired and processed using Zeiss Zen software and an ORCA-flash 4.0 LT (Hamamatsu) camera. HIF-1$\alpha$ localization and TNT formation were assessed based on confocal images: specimens were observed with a Zeiss LSM 710 confocal microscope that uses a Zeiss AXIO Observer Z1 inverted microscope stand with transmitted (HAL), UV (HBO), and laser illumination sources. Images were acquired with a Zeiss ×63 (oil) NA 1.35 objective. For all the other images, visualization and analysis were performed with ImageJ.

To quantify HIF-1$\alpha$ localization, we manually assessed the ratio of HIF-1$\alpha$ signal intensity between the nucleus and cytoplasm in over 50 cells from three different donors. The HIV infection index (total number of nuclei in HIV-stained cells divided by the total number of nuclei × 100) was quantified, as previously described

(Souriant et al, 2019). The fusion index is defined as the number of nuclei present in a multinucleated giant cell (>2 nuclei) relative to the total number of nuclei (Verollet et al, 2015). For TNT quantification, TNTs were detected and counted using F-actin and microtubule staining. Thick and thin nanotubes were quantified: thin membrane nanotubes contained only F-actin, whereas thick TNTs contained both F-actin and microtubules, as described in Souriant et al (2019).

**Flow cytometry**

Adherent cells were harvested after 15 min of incubation in trypsin–EDTA (0.05%; Gibco) and washed with PBS (Gibco). After five min of centrifugation (Eppendorf 5810R, rotor A-4-62) at 327G, pellets were resuspended in cold staining buffer (PBS, 2 mM EDTA, 0.5% FBS) with fluorophore-conjugated antibodies (Table 1). For intracellular staining, cells were fixed with 3.7% PFA for 1 h and stained with fluorophore-conjugated antibodies (Table 1) in staining buffer containing 0.15% Triton. After staining, cells were washed with cold staining buffer, centrifuged for 5 min at 379G at 4°C, and analyzed by flow cytometry using a BD LSRFortessa flow cytometer (BD Biosciences, TRI-Genotoul platform) and the associated BD FACSDiva software. Data were analyzed using FlowJo_V10 (FlowJo, LLC). For mitochondrial analysis, adherent cells were stained for 10 min at 37°C with MitoTracker or MitoSOX, according to the supplier's protocols. For glucose uptake, cells were incubated with the fluorescent glucose analog 2-(N-(7-nitrobenz-2-oxa-1,3-diazol-4-yl)-amino)-2-deoxyglucose (2-NBDG) (10 $\mu$M; Invitrogen) in PBS for 30 min. Thereafter, cells were washed, and intracellular 2-NBDG was measured by flow cytometry.

**Determination of lactate release**

Lactate production concentrations in culture media were measured using spectrophotometric lactate assay kits from Wiener (Argentina), which are based on the oxidation of lactate (Marin Franco et al, 2020). The absorbance was read using CLARIOstar Microplate Reader and its software.

**Transmission electron microscopy**

Cells were fixed in 2.5% glutaraldehyde and 2% PFA (EMS, Delta-Microscopies) dissolved in 0.1 M Sorensen buffer (pH 7.2) for 2 h at RT, and then preserved in 1% PFA dissolved in Sorensen buffer. Adherent cells were treated for 1 h with 1% aqueous uranyl acetate, then dehydrated in a graded ethanol series, and embedded in Epon. Sections were cut on a Leica Ultracut microtome, and ultrathin sections were mounted on 200-mesh Formvar-coated carbon-coated copper grids. Finally, thin sections were stained with 1% uranyl acetate and lead citrate and examined with a transmission electron microscope (JEOL JEM-1400) at 80 kV. Visualization and quantification of mitochondria were performed as described previously (Genoula et al, 2020). Images were acquired using a digital camera (Gatan Orius).

## Scanning electron microscopy

Cells were washed three times for five min in 0.2 M cacodylate buffer (pH 7.4), postfixed for 1 h in 1% (wt/vol) osmium tetroxide in 0.2 M cacodylate buffer (pH 7.4), and washed with distilled water. Samples were dehydrated through a graded ethanol series (25–100%), transferred to acetone, and subjected to critical-point drying with $CO_2$ in a Leica EM CPD300. Dried specimens were sputter-coated with 3 nm of platinum using a Leica EM MED020 evaporator and were examined and photographed with a FEI Quanta FEG 250.

## Real-time cell metabolic analysis using Seahorse

Cells ($5 \times 10^5$ cells/well) were plated after isolation on XFe24 cell culture plates (Agilent), and treated with RPMI/cmCTR/cmMTB/HF-PE or TB-PE for 1, 2, or 3 days. One hour before the assay, images of each well were captured (Incucyte) to allow normalization based on the area occupied by the cells. Then, cells were washed and replaced with DMEM (Sigma-Aldrich) supplemented with 4.5 $g$/liter d-glucose, two mM glutamine, and two mM pyruvate, followed by an incubation without $CO_2$ at 37°C for 40 min. The Mito Stress assay was performed by sequential addition of 1.5 $\mu$g/ml oligomycin (inhibitor of ATP synthesis), 0.7 $\mu$M carbonyl cyanide 4-(trifluoromethoxy) phenyl-hydrazone (FCCP, uncoupling agent), and one $\mu$M rotenone/antimycin A (inhibitors of complex I and complex III of the respiratory chain, respectively). The ATP rate test was performed by sequential addition of 1.5 $\mu$g/ml oligomycin (inhibitor of ATP synthesis) and one $\mu$M rotenone/antimycin A (inhibitors of complex I and complex III of the respiratory chain, respectively).

The ATP rate test was performed by sequential addition of 1.5 $\mu$g/ml oligomycin (inhibitor of ATP synthesis) and one $\mu$M rotenone/antimycin A (inhibitors of complex I and complex III of the respiratory chain, respectively). ATP production rates were calculated according to Agilent Seahorse XF Real-Time ATP Rate Assay Kit (User Guide #103592-100). Specifically, the glycolytic ATP production rate (GlycoATP) and the mitochondrial ATP production rate (MitoATP) were derived from the oxygen consumption rate (OCR) and extracellular acidification rate (ECAR) after oligomycin injection, using the manufacturer's proprietary algorithm. The total ATP production rate was calculated as the sum: total ATP production rate = GlycoATP production rate + MitoATP production rate. These rate values (in pmol ATP/min) were then normalized to the cell area per well, as determined by Incucyte imaging, and are expressed as pmol ATP/min in Fig 1.

The Glycolytic Rate assay test, which is specific for glycolytic acidification, was also performed, as described by the manufacturer (https://www.agilent.com/en/products/cell-analysis/glycolysis-assays-using-cell-analysis-technology). The key parameter of this assay, glycolytic proton efflux rate (GlycoPER), correlates 1:1 with lactate accumulation over time.

## Live imaging

For live imaging, 3 days after HIV-1-GFP infection, specimens were imaged on an Andor/Olympus spinning disk microscope equipped with a Yokogawa CSU-X1 scanner unit and an EMCCD camera (Andor iXon 888) under control of iQ3 software (Andor Oxford Instruments). Images were acquired with an Olympus ×60 (oil) NA 1.35 objective at 37°C for DIC and GFP (HIV) signal (one image every minute and a half).

## Transcriptomics and GSEA

The transcriptomic data from cells conditioned with cmCTR and cmMTB supernatants for 3 d were described in Dupont et al (2022) and are available under the GEO accession number GSE139511. We preprocessed the data as in the original publication and applied gene set enrichment analysis (GSEA) using the hallmark gene sets available in MSigDB (v7.5.1) to gain insights into the pathways differentially regulated between the two conditions (Subramanian et al, 2005). GSEA allows statistical testing of whether a gene set is significantly enriched in one condition relative to another, based on expression profiles. We used the following parameters: metric for ranking genes: Signal2Noise; permutation type: gene_set; number of permutations: 1,000.

## Analysis of extracellular metabolites by proton $^1$H-NMR

The culture supernatant (180 $\mu$l) of CD14$^+$ monocytes was collected at different time points (12, 20, 40, and 60 h) after treatment with the secretome of Mtb-infected macrophages. 20 $\mu$l of 10 mM (trimethylsilyl)propionic acid d4 (TSPd4) solution dissolved in D2O was added to the samples for frequency calibration and concentration measurements. The final volume of 200 $\mu$l of the resulting samples was transferred into three-mm NMR tubes. Samples were analyzed by 1H-1D NMR on a Bruker Avance III HD 800-MHz spectrometer equipped with a five-mm quadruple-resonance QCI-P (H/P-C/N/D) cryogenically cooled probe head. NMR spectra were recorded and processed using Bruker TopSpin 3.2. 1H-1D NMR spectra were acquired using a quantitative zgpr30 sequence at 280 K with 32 scans, 131k points, an acquisition time of 4 s, and a recycle delay of 8 s. Lactate and glucose consumption fluxes are shown.

## Statistical analysis

All statistical analyses were performed using GraphPad Prism 9 (GraphPad Software Inc.). Two-tailed paired or unpaired $t$ test was applied on datasets with a normal distribution (determined using the Kolmogorov–Smirnov test), whereas two-tailed Mann–Whitney (unpaired test) or Wilcoxon matched-pairs signed-rank tests were used otherwise. Bar histograms represent the mean with SD for data with a normal distribution, and the median with the interquartile range otherwise. When multiple comparisons were done, the statistical analyses used were detailed in the corresponding figure legend. $P < 0.05$ was considered as the level of statistical significance (*$P \leq 0.05$; **$P \leq 0.01$; ***$P \leq 0.001$; ****$P \leq 0.0001$).

# Supplementary Information

# Acknowledgements

We greatly acknowledge Flavie Moreau and Céline Berrone, IPBS, and Genotoul Anexplo-IPBS, for accessing the BSL3 facilities; and the Genotoul TRI-IPBS facilities for imaging and flow cytometry, in particular, Emmanuelle Näser, Serge Mazères, and Eve Pitot. We also greatly acknowledge Serge Bénichou for critical reading of the article. This work was supported by the *Centre National de la Recherche Scientifique*, *Université Paul Sabatier*, the *Institut National de la Santé et de la Recherche Médicale*, the *Agence Nationale de la Recherche* (ANR16-CE13-0005-01, ANR DFG 2020 JA-3038/2-1), the *Agence Nationale de Recherche sur le Sida et les hépatites virales (ANRS I MIE)* grant numbers: ECTZ 118551/118554, ECTZ 205320/305352, ECTZ 242543/293306, *Sidaction*, and the *Fondation pour la Recherche Médicale*. Part of this work was also supported by grants from the ANRS (ECTZ 88611 and 192763) to F Blanchet. Q Hertel is a recipient of a UM-CBS2 doctoral fellowship. We also thank the AIDS Research and Reference Reagent Program, Division of AIDS, NIAID. SC Monard and N Faivre are PhD candidates at Université Toulouse III Paul Sabatier supported by ANRS. Z Vahlas was supported by ANR.

## Author Contributions

Z Vahlas: conceptualization, data curation, formal analysis, supervision, investigation, methodology, project administration, and writing—original draft, review, and editing.
C Deyts: data curation, formal analysis, investigation, visualization, and methodology.
S Fried: data curation, formal analysis, investigation, and methodology.
M Ben Neji: data curation, formal analysis, investigation, and methodology.
M Pingret: data curation, formal analysis, investigation, visualization, and methodology.
N Faivre: data curation, formal analysis, investigation, visualization, and methodology.
SC Monard: data curation, formal analysis, investigation, and methodology.
Q Hertel: data curation, formal analysis, investigation, and methodology.
M Maio: data curation, formal analysis, investigation, and methodology.
J Barros: data curation, formal analysis, investigation, and methodology.
A Lucas: data curation, formal analysis, investigation, and methodology.
TP Vu Manh: software, formal analysis, investigation, and visualization.
M Corti: resources.
R Poincloux: resources, formal analysis, visualization, and methodology.
F Blanchet: resources and funding acquisition.
B Raynaud-Messina: supervision and investigation.
F Letisse: formal analysis, validation, visualization, and methodology.
O Neyrolles: resources, supervision, funding acquisition, project administration, and writing—review and editing.
G Lugo-Villarino: conceptualization, data curation, formal analysis, supervision, funding acquisition, validation, investigation, methodology, project administration, and writing—original draft, review, and editing.
L Balboa: conceptualization, resources, data curation, formal analysis, supervision, funding acquisition, validation, investigation, visualization, methodology, project administration, and writing—review and editing.
C Vérollet: conceptualization, resources, data curation, formal analysis, supervision, funding acquisition, validation, investigation, visualization, methodology, project administration, and writing—original draft, review, and editing.

## Conflict of Interest Statement

The authors declare that they have no conflict of interest.

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
