## [Reviewer comments · Life Science Alliance]

Glycolysis inhibition in tuberculosis-driven metabolic rewiring reduces HIV-1 spread in macrophages

Zoi Vahlas, Clara Deyts, Steven Fried, Myriam Ben Neji, Maxime Pingret, Natacha Faivre, Sarah Monard, Quentin Hertel, Mariano Maio, Joaquina Barros, alexandre Lucas, Thien-Phong Vu Mahn, Marcelo Corti, Renaud Poincloux, Fabien Blanchet, Brigitte Raynaud-Messina, Fabien Letisse, Olivier Neyrolles, Geanncarlo Lugo-Villarino, Luciana Balboa, and Christel VEROLLET

DOI: <https://doi.org/10.26508/lsa.202503333>

Corresponding author(s): Geanncarlo Lugo-Villarino, Institut de Pharmacologie et de Biologie Structurale and Christel VEROLLET, Institut de Pharmacologie et de Biologie Structurale

Review Timeline:

Submission Date:	2025-03-31
Editorial Decision:	2025-05-13
Revision Received:	2026-01-22
Editorial Decision:	2026-02-03
Revision Received:	2026-02-05
Accepted:	2026-02-06

Scientific Editor: Tim Fessenden

Transaction Report:

Please note that the manuscript was reviewed at *Review Commons* and these reports were taken into account in the decision-making process at *Life Science Alliance*.

Reviews

Review #1

In the manuscript entitled "Inhibition of glycolysis in tuberculosis-mediated metabolic rewiring reduces HIV-1 spread across macrophages", Vahlas and colleagues investigated the hypothesis that Mtb interferes with HIV-1 infection of human macrophages, as they represent a common target cell type. In particular, they observed that a conditioned medium generated from Mtb-infected macrophages (Mtb-CM) induces tunneling nanotubes (TNT) in HIV-infected macrophages thereby facilitating viral spreading. At the same time, Mtb-CM induced a glycolytic pathway leading to ATP accumulation in HIV-infected macrophages, an essential pathway for TNT induction whereas pharmacological interference with such a metabolic switch resulted in a reduced viral production.

Experimental approach: primary human monocytes differentiated into monocyte-derived macrophages (MDM) in the presence of a TB-dominated microenvironment (Mtb-CM). The intracellular rate of ATP production was evaluated by the Seahorse technology at day 3 of MDM differentiation. The measurements of basal extracellular acidification rate (ECAR) and basal oxygen consumption rate (OCR) were used to calculate ATP production rate from glycolysis (GlycoATP) and mitochondrial OXPHOS (MitoATP).

This is a well-conducted, innovative study exploring the interaction of two main human pathogens, i.e. Mtb and HIV, sharing macrophages as common target cell. The manuscript is clearly written and the conclusions and hypotheses are supported by experimental evidence. I have two general points that I encourage the authors to address.

1. As mentioned in the Discussion, macrophage infection by HIV is characterized by the accumulation of preformed, infectious virions in VCC (Virus Containing Compartments) that can be pharmacologically modulated both in terms of accumulation and rapid release in the absence of cell cytopathicity. Although the modulation of VCC was not the objective of the present study, it would be important to discuss their role and their potential modulation by Mtb and/or metabolic modifications, if known.
2. Understanding the purpose of using a VSV-g based infection system, nonetheless it would be important to know whether metabolic modulation does affect CD4 and CCR5 expression on MDM and its consequence for their susceptibility to HIV infection, in addition to the effects on TNT formation and viral transfer between cells.

****Specific points:****

1. "TB-PE" (pleural effusion) is neither specified in the Results nor in the Methods sections.
2. Figure 3A does not seem to display cell viability, but rather HIV Gag expression by IFA. Furthermore, Figure 3C indicates Gag expression, not "HIV infection" (see page 8, Results).

2. Significance:

The paper addresses a poorly explored area, i.e. the interaction of Mtb and HIV during infection of macrophages. The authors focused on a specific aspect of such an interaction (i.e. the modulation of nanotubes formation and transfer of virions to target cells), but their results can be extrapolated in a broader context, particularly if the authors will be willing to address my general questions.

Although specific in its experimental approach, the implication of the study will be of interest to a general audience.

3. How much time do you estimate the authors will need to complete the suggested revisions:

Less than 1 month

----- Review #2

The current work is based on previous observations that the abundance of lung macrophages is augmented in NHPs with active TB and exacerbated in those coinfecting with SIV (Dupont et al., 2022; Dupont et al., 2020; Souriant et al., 2019). Further work with these TB-induced immunomodulatory macrophages demonstrated an increased susceptibility to HIV-1 replication and spread via the formation of tunneling nanotubes (TNTs), (Souriant et al., 2019).

In the present manuscript, the authors connected these findings with the metabolic state of macrophages (glycolysis vs OXPHOS). Using a range of metabolic inhibitors coupled with seahorse assays and microscopy confirmed the role of Mtb-induced glycolytic shift in inducing the formation of TNTs and the spread of HIV. The work is well-planned and executed. However, the study is mainly correlative without any molecular insights. The knowledge generated is important and valuable for future studies to understand the molecular players in regulating immunometabolism during HIV-TB coinfection.

****Major Comments****

There are conflicting reports about Mtb's impact on macrophage ECAR and OXPHOS, which authors have acknowledged. Therefore, including OCR and ECAR plots along with the glycoATP and MitoATP data will be useful. Similarly, OCR/ECAR plots without any conditioned medium should be included to clarify the role of Mtb infection on OCR/ECAR.

Fig 2G image is not convincing. While HIF1 alpha seems more in the nucleus, the overall morphology of the cell is more compact. Additional verification is needed. Furthermore, genetic evidence is required in order to confirm if HIF1 alpha is the primary regulator of glycolytic shift by cmMTB/PE-TB, leading to more HIV dissemination by the TNT formation.

Also, the authors have used only one tool to measure HIV levels -microscopy. While important, another method for verifying findings is needed. This is important as the effect of inhibitors (UK5099) is marginal.

Authors have used oxamate to inhibit glycolysis. Inhibition of LDH could lead to inhibition of NAD/NADH regeneration, thereby slowing down glycolysis. However, lack of lactate could have wide-ranging influence on cells as lactate could regulate several post-translational modifications, including lactylation. While the authors argued against using 2-DG, several findings confirm the glycolysis inhibitory potential of 2-DG when infected with Mtb. This should be included.

A standard glycolytic function test (glucose, oligomycin and 2-DG injection) should be performed to assess the effect of TB-PE and cmMTB on the macrophages directly.

Depriving glucose is not the best way to show the effect of glucose on HIV infection and MGC formation, as it can affect other aspects of cellular physiology, such as redox and bioenergetics. Instead, the use of galactose in place of glucose would generate ATP only by OXPHOS. Some key experiments should be repeated using galactose as a sole C source.

UK5099 and oxamate nuclei seem smaller and less bright compared to the control. Images between control and UK5099 appear marginally different (non-significant).

The overall impact of the study is limited as the authors provide no evidence on the mechanism of how glycolysis induces TNT formation, which needs to be more characterized.

****Minor comments:****

The manuscript does not clearly show how the total ATP was calculated from the ATP rate assay.

In figure 1 (and everywhere else) the units on the y-axis should be corrected to [pmol/min] instead of pmol and the Seahorse profiles should mention whether the axis represents OCR or ECAR.

The authors have called the macrophages highly glycolytic in first set of results which is misleading. Although the glycoATP contribution is increasing, overall ATP production is still majorly through oxidative phosphorylation (70% vs 25%).

Fig 3: Why does the HIV gag protein signal appear as irregular large spots?

****Referees cross-commenting****

I agree with the reviewer# 1 assessment. However, i feel that mechanistically paper could be improved and by performing more experiments.

2. Significance:

The knowledge generated is important and valuable for future studies to understand the molecular players in regulating immunometabolism during HIV-TB coinfection.

3. How much time do you estimate the authors will need to complete the suggested revisions:

Between 1 and 3 months.

We thank the reviewers for their insightful comments, and we address all their comments in the detailed point-by-point responses provided below.

Reviewer #1

Evidence, reproducibility and clarity

In the manuscript entitled "Inhibition of glycolysis in tuberculosis-mediated metabolic rewiring reduces HIV-1 spread across macrophages", Vahlas and colleagues investigated the hypothesis that Mtb interferes with HIV-1 infection of human macrophages, as they represent a common target cell type. In particular, they observed that a conditioned medium generated from Mtb-infected macrophages (Mtb-CM) induces tunneling nanotubes (TNT) in HIV-infected macrophages thereby facilitating viral spreading. At the same time, Mtb-CM induced a glycolytic pathway leading to ATP accumulation in HIV-infected macrophages, an essential pathway for TNT induction whereas pharmacological interference with such a metabolic switch resulted in a reduced viral production.

Experimental approach: primary human monocytes differentiated into monocyte-derived macrophages (MDM) in the presence of a TB-dominated microenvironment (Mtb-CM). The intracellular rate of ATP production was evaluated by the Seahorse technology at day 3 of MDM differentiation. The measurements of basal extracellular acidification rate (ECAR) and basal oxygen consumption rate (OCR) were used to calculate ATP production rate from glycolysis (GlycoATP) and mitochondrial OXPHOS (MitoATP).

This is a well-conducted, innovative study exploring the interaction of two main human pathogens, i.e. Mtb and HIV, sharing macrophages as common target cell. The manuscript is clearly written and the conclusions and hypotheses are supported by experimental evidence. I have two general points that I encourage the authors to address.

We thank the Reviewer for his/her valuable comments and address all provided comments below.

1. As mentioned in the Discussion, macrophage infection by HIV is characterized by the accumulation of preformed, infectious virions in VCC (Virus Containing Compartments) that can be pharmacologically modulated both in terms of accumulation and rapid release in the absence of cell cytopathicity. Although the modulation of VCC was not the objective of the present study, it would be important to discuss their role and their potential modulation by Mtb and/or metabolic modifications, if known.

In the discussion, we mentioned that "In HIV-1 infected macrophages, ATP is also vital for the release of particles from virus-containing compartments (Graziano et al., 2015)". Graziano et al. (PMID 26056317) showed that extracellular ATP favors the release of virions actively accumulating within the VCC of infected macrophages through its interaction with the P2X7 receptor. This study will be discussed more in detail in the revised version of the manuscript.

In addition, we fully agree with the reviewer that exploring potential modifications in the formation of virus containing compartments (VCC) following Mtb infection, CmMTB treatment or metabolic alterations is highly relevant. Importantly, VCCs are specific compartments in infected macrophages where new virions are generated and protected from the immune system and antiretroviral therapies. Interestingly, Siglec-1 was shown to be involved in VCC formation in infected macrophages (Jason E Hammonds et al., 2017; PMID 28129379), and we demonstrated that the level of expression of this lectin is increased in CmMTB-treated cells (Dupont et al., PMID: 32223897). We propose to perform new experiments during the revision process to look whether the formation of VCC is disturbed in CmMTB-treated macrophages upon HIV-1 infection, using the tetraspanin CD81 and/or Siglec-1 along with HIV-Gag to assess VCC formation (as in Reviewer Figure 1).

Reviewer Figure 1: VCC formation in multinucleated HIV-1 infected macrophages. Human macrophages were infected with HIV-1 (NLAd8-VSVG, 3 days) and stained with HIV-gag and CD81 to stain the VCC.

2. Understanding the purpose of using a VSV-g based infection system, nonetheless it would be important to know whether metabolic modulation does affect CD4 and CCR5 expression on MDM and its consequence for their susceptibility to HIV infection, in addition to the effects on TNT formation and viral transfer between cells.

We appreciate this comment. The reviewer correctly understands that we used VSVG pseudotyped virus in this study to eliminate the effect of metabolic modulation on the expression of HIV entry receptors and potentially on virus entry. It has been previously demonstrated in CD4 T cells that the nutrient modulation does not affect HIV entry when the Blam-Vpr assay is used (Clerc et al., 2019, PMID 32373781, supplemental Figure 6).

In addition, as demonstrated in our earlier work (Souriant et al. Cell Reports, 2019), CmMTB treatment increases the levels of both CD4 and CCR5 on the surface of macrophages. However, it does not impact HIV entry, as shown using the same Blam-Vpr assay. Therefore, the exacerbation of HIV-1 infection in the TB-environment is not a consequence of increased viral entry. This will be clarified in the revised version of the manuscript.

As suggested by the reviewer, we will also conduct new experiments during the revision process. Specifically, we will assess the levels of entry receptors using flow cytometry analysis and measure virus

entry using the Blam-Vpr fusion assay in CmMTB-treated cells, with or without Oxamate treatment (to inhibit glycolysis).

Specific points:

1. "TB-PE" (pleural effusion) is neither specified in the Results nor in the Methods sections.

We thank the reviewer for pointing out this omission. TB-PE refers to pleural effusions from TB patients, a term we had previously defined only in the introduction and figure legends. We will ensure that this definition is explicitly stated in the Result and Methods sections of the revised manuscript.

2. Figure 3A does not seem to display cell viability, but rather HIV Gag expression by IFA.

Indeed, there is an error in the text regarding cell viability. Cell viability following drug treatments was assessed by flow cytometry, as shown in Figure S2C. In Figure 3A, we included nuclear staining (in addition to HIV Gag) to confirm that cell density is not affected. This will be corrected in the revised manuscript. Additionally, we will perform F-actin staining to evaluate cell morphology and further confirm that all key parameters, i.e., viability, cell density, and cell morphology, are unaffected by the drugs used in Figure 3.

Furthermore, Figure 3C indicates Gag expression, not "HIV infection" (see page 8, Results).

We thank the reviewer for helping us to clarify this issue. In Figure 3C, the term "infection index" refers to the percentage of HIV Gag-positive cells resulting from productive infection. This is calculated as the total number of nuclei in HIV Gag-stained cells divided by the total number of nuclei, multiplied by 100, as described in the Methods section.

We have previously used this method to estimate the HIV infection rate in our published studies (Souriant et al., 2019; Dupont et al., 2020; Mascarau et al., 2023). To further improve the clarity and interpretation of the figure, we will include a clear definition of the infection index in the figure legend in the revised version of the manuscript.

Significance

The paper addresses a poorly explored area, i.e. the interaction of Mtb and HIV during infection of macrophages. The authors focused on a specific aspect of such an interaction (i.e, the modulation of nanotubes formation and transfer of virions to target cells), but their results can be extrapolated in a broader context, particularly if the authors will be willing to address my general questions. Although specific in its experimental approach, the implication of the study will be of interest to a general audience.

We appreciate this positive comment.

Reviewer #2

Evidence, reproducibility and clarity

The current work is based on previous observations that the abundance of lung macrophages is augmented in NHPs with active TB and exacerbated in those coinfecting with SIV (Dupont et al., 2022; Dupont et al., 2020; Souriant et al., 2019). Further work with these TB-induced immunomodulatory macrophages demonstrated an increased susceptibility to HIV-1 replication and spread via the formation of tunneling nanotubes (TNTs), (Souriant et al., 2019). In the present manuscript, the authors connected these findings with the metabolic state of macrophages (glycolysis vs OXPHOS). Using a range of metabolic inhibitors coupled with seahorse assays and microscopy confirmed the role of Mtb-induced glycolytic shift in inducing the formation of TNTs and the spread of HIV. The work is well-planned and executed. However, the study is mainly correlative without any molecular insights. The knowledge generated is important and valuable for future studies to understand the molecular players in regulating immunometabolism during HIV-TB coinfection.

We thank the Reviewer for his/her valuable comments, and we address all provided comments below.

Major Comments:

There are conflicting reports about Mtb's impact on macrophage ECAR and OXPHOS, which authors have acknowledged. Therefore, including OCR and ECAR plots along with the glycoATP and MitoATP data will be useful. Similarly, OCR/ECAR plots without any conditioned medium should be included to clarify the role of Mtb infection on OCR/ECAR.

In this manuscript, we evaluated the intracellular rate of ATP production in macrophages (day 3 of differentiation) treated with either cmCTR or cmMTB using Seahorse technology. Measurements of extracellular acidification rate (ECAR) and oxygen consumption rate (OCR), both before and after the addition of oligomycin (an ATP synthase inhibitor), were used to calculate the contributions of glycolysis (GlycoATP, Figure 1B) and mitochondrial OXPHOS (MitoATP, Figure S1C) to total ATP production (Figure 1A).

We agree with the reviewer that displaying basal OCR/ECAR plots (bioenergetic profiles) would help characterize the overall energy phenotypes of macrophages. These graphs will be prepared and included in Figure S1. Furthermore, we will enhance the discussion and interpretation of these findings in the Results section of the revised manuscript.

As suggested, we will also assess ATP production using Seahorse technology for control cells (day 3 differentiated in RPMI) and provide OCR/ECAR plots for these new experiments.

Fig 2G image is not convincing. While HIF1 alpha seems more in the nucleus, the overall morphology of the cell is more compact. Additional verification is needed.

Regarding the specific comment on Fig. 2G, the reviewer is correct that the morphology of CmMTB-treated cells differs from that of CmCTR-treated cells. We have previously shown that CmMTB-treated macrophages display an M(IL-10) phenotype, characterized by a CD16+CD163+MerTK+PD-L1+ signature, morphological changes (cells appear rounder and form more TNTs), nuclear translocation of phosphorylated STAT3, and increased susceptibility to Mtb or HIV-1 infection (Dupont et al., 2022; Dupont et al., 2020; Lastrucci et al., 2015; Souriant et al., 2019).

As shown in Figure 2H, HIF1- α is predominantly cytoplasmic in most control cells, whereas an increased number of cells with nuclear HIF-1 α staining were observed in CmMTB-treated cells. To quantify this observation, we manually assessed the ratio of HIF-1 α signal intensity between the nucleus and cytoplasm in over 50 cells from three different donors. This methodology was not adequately explained in the Methods section and will be clarified in the revised manuscript. We also propose to include more representative images of HIF-1 α -stained cells to support these findings.

Furthermore, genetic evidence is required in order to confirm if HIF1 alpha is the primary regulator of glycolytic shift by cmMTB/PE-TB, leading to more HIV dissemination by the TNT formation.

We fully agree that further experiments are essential to formally demonstrate that HIF-1 α activation is responsible for the observed increase in HIV-1 infection and TNT formation in CmMTB-treated cells. To address this hypothesis, we propose conducting key experiments during the revision process

We will first use pharmacological approaches to modulate HIF-1 α levels, as described in our recent publication (Maio et al., eLife, PMID 38922679). Specifically, we will test the HIF-1 α inhibitor PX-478 as well as dimethylxalylglycine (DMOG), a compound that stabilizes HIF-1 α expression. These drugs will be applied 24h prior to HIV-1 infection in CmMTB-treated cells, and we will quantify HIV-1 infection and TNT formation on day 6 using immunofluorescence (IF).

In parallel, though technically challenging, we will attempt to reduce HIF-1 α expression (and consequently its activity) in primary human monocytes using a siRNA-mediated depletion approach. This method has been successfully employed in our previous studies to target STAT3, STAT1 and Siglec-1 (Dupont et al., 2020; Lastrucci et al., 2015; Dupont et al., 2022). Under these conditions, we will measure HIV-1 infection and TNT formation on day 6 by IF.

Also, the authors have used only one tool to measure HIV levels -microscopy. While important, another method for verifying findings is needed. This is important as the effect of inhibitors (UK5099) is marginal.

In the present manuscript, we assess HIV-1 infection levels using two methods: microscopy (Figure 3 and 4I) and flow cytometry (Figure S2H-I). To address the reviewer's comment, we propose to complement our current analysis of HIV-1 infection by evaluating HIV-1 replication through the measurement of HIV-p24 release in the supernatant of CmMTB-treated macrophages following drug treatments, as previously performed (Dupont et al., 2020; Souriant et al., 2019; Dupont et al., 2022; Mascarau et al., 2024; Raynaud-Messina et al., 2018).

Regarding the slight increase of HIV-1 infection (Gag expression by IF, Figure 3A) upon UK5099 treatment, we appreciate the reviewer's valuable observation. Enhancing glycolysis levels remains a considerable challenge in studies targeting metabolic pathways, as most approaches focus on inhibiting glycolysis. However, in our study, the effect UK5099 on HIV-1 infection is reproducible and statistically significant, as demonstrated by analyzes of data from more than ten donors using IF (Figure 3C) and eight donors by flow cytometry (Figure S2H-I).

We acknowledge that the specific image provided in Fig. 3A for the UK5099 condition may not be the most representative and could cause confusion. To address this, we will replace the current image with a more representative one in the revised version of the manuscript.

Authors have used oxamate to inhibit glycolysis. Inhibition of LDH could lead to inhibition of NAD/NADH regeneration, thereby slowing down glycolysis. However, lack of lactate could have wide-ranging influence on cells as lactate could regulate several post-translational modifications, including lactylation. While the authors argued against using 2-DG, several findings confirm the glycolysis inhibitory potential of 2-DG when infected with Mtb. This should be included.

We understand the reviewer's comment regarding the glucose analog 2-DG, which is widely used to inhibit glycolysis. Notably, recent studies have used it to show that glycolytic activity is critical for reactivating HIV-1 in macrophage reservoirs (Real et al., 2022, PMID 36220814).

In our study, we did not initially use 2-DG because it also inhibits glucose contribution to OXPHOS, making it challenging to distinguish between the roles of glycolysis and OXPHOS in macrophages (Wang et al., Cell Metabolism, PMID 30184486). Unlike Oxamate or GSK 2837, which specifically target LDHA, 2-DG does not exclusively affect glycolysis. Furthermore, inhibiting glucose metabolism with 2-DG is expected to yield similar results to glucose deprivation, as demonstrated in Figures 3H-K.

To address this, we propose conducting the suggested experiments using 2-DG in CmMTB-treated macrophages during the revision process. This will allow to assess their susceptibility to HIV-1 under this treatment. We will subsequently discuss the effects of 2-DG and integrate these results into the revised version of the manuscript.

A standard glycolytic function test (glucose, oligomycin and 2-DG injection) should be performed to assess the effect of TB-PE and cmMTB on the macrophages directly.

We appreciate the reviewer's comment and will address it by testing the ability of CmMTB to alter the glycolytic activity of macrophages using the Seahorse Glycolytic Rate Assay. This assay, a refined version of the classical Seahorse Glycolysis Stress Test (see <https://www.agilent.com/en/products/cell-analysis/glycolysis-assays-using-cell-analysis-technology>), relies on an algorithm that generates the Proton Efflux Rate (PER), providing a robust quantitative measurement of glycolytic function. PER is directly correlated with lactate accumulation, enabling us to calculate glycolytic parameters that will complement our existing assays aimed at characterizing the glycolytic pathway in CmMTB-treated macrophages. We plan to perform these measurements and include the results in Figure 2.

Depriving glucose is not the best way to show the effect of glucose on HIV infection and MGC formation, as it can affect other aspects of cellular physiology, such as redox and bioenergetics. Instead, the use of galactose in place of glucose would generate ATP only by OXPHOS. Some key experiments should be repeated using galactose as a sole C source.

We agree with this comment. In M2 macrophages, it has been shown that both glucose deprivation (as demonstrated in this study, Figure 3H-K) and glucose substitution with galactose (Wang et al., Cell Metabolism, PMID 30184486) effectively suppress glycolytic activity. Galactose must first be metabolized by the Leloir pathway before entering glycolysis, resulting in a significant reduction in glycolytic flux.

As suggested by the reviewer, we will complement our study by using galactose as the carbon source instead of glucose in a new set of experiments during the revision process.

UK5099 and oxamate nuclei seem smaller and less bright compared to the control. Images between control and UK5099 appear marginally different (non-significant).

Figure 3A may not clearly convey that the nuclei are unaffected by the treatment. To address this, we will adjust the images, particularly the DAPI staining settings, to ensure accurate interpretation.

Regarding the slight effect of UK5099 treatment on Gag expression (infection index), as discussed above, this effect is reproducible and significant. We will replace the current image in Figure 3A with a more representative one.

The overall impact of the study is limited as the authors provide no evidence on the mechanism of how glycolysis induces TNT formation, which needs to be more characterized.

We fully agree that understanding how glycolysis induces tunneling nanotubes (TNTs) is a crucial and challenging question. This challenge stems from the incomplete understanding of the molecular mechanisms underlying TNT formation and the contradictory results reported across different cell types.

In our study, we demonstrated that inhibiting glycolysis—using Oxamate, GSK, or glucose deprivation—reduces TNT formation, whereas promoting glycolysis with UK5099 enhances their formation. We discuss in the manuscript that glycolysis likely provides the energy required for actin cytoskeletal rearrangements, which are essential for TNT formation.

Moreover, ATP plays a critical role in supporting cellular functions depending on actin remodeling, such as cell migration and the epithelial-to-mesenchymal transition (DeWane et al., 2021, PMID33558441).

To try to investigate the molecular mechanisms underlying TNT formation in our model, we propose the following experiments during the revision process:

- HIF1- α and TNT formation: IF staining of HIF1- α will be performed to correlate TNT formation with the level of HIF1- α nuclear translocation (as quantified in Figure 2I). This experiment aims to demonstrate a link between HIF1- α activation and TNT formation.

- Effect of HIF1- α inhibition: TNT formation will be quantified upon inhibiting HIF1- α activity using pharmacological approaches and/or siRNA-mediated gene silencing in HIV-1-infected CmMTB-treated cells.
- GLUT-1 focalization and TNT formation: To establish a connection between glycolysis and TNT formation, we will localize the primary glucose transporter GLUT-1 in relation to TNTs in CmMTB-treated macrophages. This approach builds on previous work on microvilli, which are F-actin structures with similarities to TNTs (Hexige et al., 2015, PMID: 25561062). Confocal or super-resolution microscopy will be employed to determine whether GLUT-1 accumulates at specific TNT sites.

Through these experiments, we aim to provide deeper insights into the role of glycolysis in TNT formation.

Minor comments:

The manuscript does not clearly show how the total ATP was calculated from the ATP rate assay.

We will ensure that the method for calculating total ATP is explicitly described in the Methods section of the revised manuscript.

In figure 1 (and everywhere else) the units on the y-axis should be corrected to [pmol/min] instead of pmol and the Seahorse profiles should mention whether the axis represents OCR or ECAR.

The reviewer is correct. The axes in the relevant figures for ATP rate results (Figure 1A, B, C, D and Figure S1A, B, C) will be revised in the updated version of the manuscript.

The authors have called the macrophages highly glycolytic in first set of results which is misleading. Although the glycoATP contribution is increasing, overall ATP production is still majorly through oxidative phosphorylation (70% vs 25%).

We fully agree with the reviewer's comment. As mentioned in the Result section "Approximately 90% of ATP production in macrophages differentiated with cmCTR came from OXPHOS; this parameter was reduced to 70% when conditioned with cmMTB (Figure 1E-F)." CmMTB and TB-PE drive macrophages toward an M2/M(IL-10) phenotype (Lastrucci et al. 2015), and based on the extensive literature on metabolism of anti-inflammatory M2 macrophages, this phenotype primarily relies on OXPHOS and fatty acid oxidation (for review see Biswas and Mantovani, Cell Metabolism, 2012).

It is therefore logical that overall ATP production in these cells remains predominantly through OXPHOS. However, we observe a significant decrease in OXPHOS activity following CmMTB treatment, alongside a marked increase in glycolysis (Figure 1).

Referring to CmMTB-treated macrophages as highly glycolytic was inaccurate, indeed, and this terminology will be corrected, with a clearer explanation provided in the revised manuscript.

Fig 3: Why does the HIV gag protein signal appear as irregular large spots?

In Figure 3A, the resolution used is sufficient to quantify the number of cells positive for HIV Gag (and thus the infection index). However, it does not allow for detailed examination of the intracellular localization of Gag as “spots”. The reviewer is correct that, within macrophages, the Gag signal often appears as large and intense cytoplasmic “spots” corresponding to the VCC, as illustrated in Reviewer Figure 1 in response to Reviewer 1.

Referees cross-commenting:

I agree with the reviewer# 1 assessment. However, I feel that mechanistically paper could be improved and by performing more experiments.

We fully agree that additional experiments are essential to improve the manuscript. We will address all comments and perform the experiments suggested by Reviewer 2, particularly to better characterize the metabolic state of our cells, provide evidence for the role of glycolysis in HIV-1 exacerbation, and further elucidate the mechanism by which glycolysis induces TNT formation.

Significance

The knowledge generated is important and valuable for future studies to understand the molecular players in regulating immunometabolism during HIV-TB coinfection.

We appreciate this positive comment.

May 13, 2025

Re: Life Science Alliance manuscript #LSA-2025-03333-T

Dr. Geanncarlo Lugo-Villarino
Institut de Pharmacologie et de Biologie Structurale
Tuberculosis and Infection Biology
205 Route De Narbonne
Toulouse, Haute-Garonne 31077
France

Dear Dr. Lugo-Villarino,

Thank you for transferring your manuscript entitled "Glycolysis inhibition in tuberculosis-driven metabolic rewiring reduces HIV-1 spread in macrophages" to Life Science Alliance from Review Commons.

This manuscript was received without the rebuttal template that Review Commons offers, which is needed to clearly indicate the nature of the revision. This manuscript was invited by the previous Executive Editor at LSA, who has now moved on. Due to these unusual circumstances this work was returned to the original reviewers at Review Commons in its present form. We sincerely regret this misstep.

As you will see below, Reviewer 2 remarks that their comments have not been addressed. We contacted this reviewer and they agreed to assess a properly revised manuscript once you have prepared it. Please disregard their comments below.

As you prepare the revision, please correct an apparent error in Figure 3, in which panels J and K appear to be identical. To upload the revised version of your manuscript, please log in to your account: <https://lsa.msubmit.net/cgi-bin/main.plex>

Please note that strong support from the referees on the revised version is needed for acceptance.

Thank you for this interesting contribution to Life Science Alliance. We are looking forward to receiving your revised manuscript.

Sincerely,

-- Summary blurb (enter in submission system): A short text summarizing in a single sentence the study (max. 200 characters including spaces). This text is used in conjunction with the titles of papers, hence should be informative and complementary to the title and running title. It should describe the context and significance of the findings for a general readership; it should be

written in the present tense and refer to the work in the third person. Author names should not be mentioned.

B. MANUSCRIPT ORGANIZATION AND FORMATTING:

Reviewer #1 (Comments to the Authors (Required)):

the revised manuscript has adequately addressed my initial concerns and it has endorsed my suggestions.

Reviewer #2 (Comments to the Authors (Required)):

I have reviewed the manuscript and my previous comments in the Review Commons. I have also checked the authors' response letter. Unfortunately, while the authors indicated in the response letter that they would address my comments and provide a detailed description of how they would be revising the manuscript, none of my major comments were addressed by the authors. I stand by my previous comments. However, if the other reviewers are satisfied with the revised manuscript, I have no issues.

Reviewer #1:

The revised manuscript has adequately addressed my initial concerns and it has endorsed my suggestions.

Reviewer #2:

I have reviewed the manuscript and my previous comments in the Review Commons. I have also checked the authors' response letter. Unfortunately, while the authors indicated in the response letter that they would address my comments and provide a detailed description of how they would be revising the manuscript, none of my major comments were addressed by the authors. I stand by my previous comments. *However, if the other reviewers are satisfied with the revised manuscript, I have no issues.*

We would like to express our sincere gratitude to the reviewers for their valuable comments and to the editor for giving us the opportunity to revise the manuscript. Our detailed point-by-point response below addresses all the reviewers' comments.

In short, we addressed the main concerns raised by the Reviewer 2 by (i) performing new Seahorse experiments including controls, (ii) changing some images for more representative ones, (iii) showing Glut-1 accumulation at the tip of TNTs, and (iv) performing new experiments with drugs (2-DG), galactose complementation and siRNA against HIF-1 α to strongly confirm the role of glycolysis in HIV exacerbation.

We prepared a revised version of the manuscript with changes highlighted in yellow. As you will see, most of the Figures and Supplemental Figures (including an entirely new Figure, Figure 5) have been modified in accordance with the reviewers' comments. Of note, the order of some authors changed for their contribution during the revision process (M. Ben Neji), and four new contributing authors were added for their strong help with siRNA experiments for the new Figure 5 (C. Deyts), Seahorse analysis (S. Fried), and microscopy images of GLUT-1 (M. Pingret), following reviewers' comments. We also provide an additional figure for the reviewers' concern only (Fig. R1).

Reviewer #2 “Review Commons” comments:**Evidence, reproducibility and clarity**

The current work is based on previous observations that the abundance of lung macrophages is augmented in NHPs with active TB and exacerbated in those coinfecting with SIV (Dupont et al., 2022; Dupont et al., 2020; Souriant et al., 2019). Further work with these TB-induced immunomodulatory macrophages demonstrated an increased susceptibility to HIV-1 replication and spread via the formation of tunneling nanotubes (TNTs), (Souriant et al., 2019). In the present manuscript, the authors connected these findings with the metabolic state of

macrophages (glycolysis vs. OXPHOS). Using a range of metabolic inhibitors coupled with Seahorse assays and microscopy confirmed the role of Mtb-induced glycolytic shift in inducing the formation of TNTs and the spread of HIV. The work is well-planned and executed. However, the study is mainly correlative without any molecular insights. The knowledge generated is important and valuable for future studies to understand the molecular players in regulating immunometabolism during HIV-TB coinfection.

We thank the Reviewer for his valuable remarks and address all comments below.

Major Comments:

There are conflicting reports about Mtb's impact on macrophage ECAR and OXPHOS, which authors have acknowledged. Therefore, including OCR and ECAR plots along with the glycoATP and MitoATP data will be useful. Similarly, OCR/ECAR plots without any conditioned medium should be included to clarify the role of Mtb infection on OCR/ECAR.

Indeed, some reports about how Mtb impacts macrophage metabolism contradict each other, as we reported in the introduction page 4, line 25-30: "Different metabolic pathways are prevalent in macrophages depending on their ontogeny, the state of TB (active or latent) and the virulence of the pathogen (live or irradiated Mtb); these states can be remarkably reversible depending on environmental cues (Beste et al., 2013; Cano-Muniz et al., 2018; Cumming et al., 2018; de Carvalho et al., 2010; Llibre et al., 2021; Pandey and Sasseti, 2008; Zimmermann et al., 2017). Therefore, results about how Mtb impacts macrophage metabolism are conflicting."

Here, we evaluated the intracellular ATP production rate in macrophages (day 3 of differentiation) treated with either cmCTR or cmMTB using Seahorse technology. Measurements of extracellular acidification rate (ECAR) and oxygen consumption rate (OCR), both before and after the addition of oligomycin (an ATP synthase inhibitor), were used to calculate the contributions of glycolysis (GlycoATP, Figure 1B) and mitochondrial OXPHOS (MitoATP, Figure S1C) to total ATP production (Figure 1A).

We agree with the reviewer that displaying basal OCR/ECAR plots (bioenergetic profiles) would help characterize macrophage energy phenotypes, and that comparing the metabolic profile of control cells (day 3 differentiated in RPMI) with the cmCTR condition is key. We thus performed these experiments, and the graph is shown in Figure S1A. Furthermore, these findings are largely addressed in the discussion section of the revised manuscript, see Page 10, line 20-44.

Fig 2G image is not convincing. While HIF1 alpha seems more in the nucleus, the overall morphology of the cell is more compact. Additional verification is needed.

Regarding the specific comment on Fig. 2G, the reviewer is correct that the morphology of CmMTB-treated cells differs from that of CmCTR-treated cells. We have previously shown that CmMTB-treated macrophages display an M(IL-10) phenotype, characterized by a CD16+CD163+MerTK+PD-L1+ signature, morphological changes (cells appear rounder and form more TNTs), nuclear translocation of phosphorylated STAT3, and increased susceptibility to Mtb or HIV-1 infection (Dupont et al., 2022;

Dupont et al., 2020; Lastrucci et al., 2015; Souriant et al., 2019). In response to the reviewer's concern, we have now changed the images (now Figure 2I) to more convincing ones.

As shown in Figure 2I, HIF1- α is predominantly cytoplasmic in most control cells, whereas an increased number of cells with nuclear HIF-1 α staining were observed in CmMTB-treated cells. To quantify this observation, we manually assessed the ratio of HIF-1 α signal intensity between the nucleus and cytoplasm in over 50 cells from three different donors. We agree that this methodology was not adequately explained in the Methods section and has now been clarified in the revised manuscript (Page 17, lines 38-39).

Furthermore, genetic evidence is required in order to confirm if HIF1 alpha is the primary regulator of glycolytic shift by cmMTB/PE-TB, leading to more HIV dissemination by the TNT formation.

We fully agree that further experiments are essential to formally demonstrate that HIF-1 α activation is responsible for the observed increase in HIV-1 infection and TNT formation in CmMTB-treated cells. To address this hypothesis, we conducted key experiments during the revision process. While technically challenging, we succeeded in reducing HIF-1 α expression (and, consequently, its activity) in primary human monocytes using siRNA-mediated gene depletion. The results are now included in Figure 5 and Supplementary Figure 5, and commented on in the results section (Page 9, lines 15-41) and in the discussion (Page 13, lines 14-31 and 34-39). We showed that inhibiting HIF-1 α expression reduces HIV-1 infection and MGC formation in CmMTB-treated macrophages.

Also, the authors have used only one tool to measure HIV levels -microscopy. While important, another method for verifying findings is needed. This is important as the effect of inhibitors (UK5099) is marginal.

In the present manuscript, we assess HIV-1 infection levels using two methods: microscopy (Figures 3 and 4) and flow cytometry (new Figure S3I-J). Microscopy is the most accurate and provides many parameters related to HIV-1 infection, such as the number of productively infected cells (p24-positive cells) and the formation of MGC. We have shown in the past that these parameters always correlate with HIV production, as assessed by HIV-p24 release in the supernatant of macrophages (Dupont et al., 2020; Souriant et al., 2019; Dupont et al., 2022; Mascarau et al., 2024; Raynaud-Messina et al., 2018). Here, the timing of our experiments (day 3 post-HIV-1 infection) and the use of a VSVG pseudotyped virus do not allow us to distinguish between the virus used for infection and that used for virus production.

Regarding the slight increase in HIV-1 infection upon UK5099 treatment (Gag expression by IF, Figure 3A), we appreciate the reviewer's valuable observation. Enhancing glycolysis levels remains a considerable challenge in studies targeting metabolic pathways, as most approaches focus on inhibiting glycolysis. However, in our study, the effect of UK5099 on HIV-1 infection is reproducible and statistically significant, as demonstrated by analyses of data from more than ten donors using IF (Figure 3C) and eight donors by flow cytometry (new Figure S3I-J). In any case, we acknowledge that the previous image in Fig. 3A for the UK5099 condition was not the most representative and could have caused confusion. We have changed this image; please see the new Figure 3A (UK5099 condition).

Authors have used oxamate to inhibit glycolysis. Inhibition of LDH could lead to inhibition of NAD/NADH regeneration, thereby slowing down glycolysis. However, lack of lactate could have wide-ranging influence on cells as lactate could regulate several post-translational modifications, including lactylation. While the authors argued against using 2-DG, several findings confirm the glycolysis inhibitory potential of 2-DG when infected with Mtb. This should be included.

We understand the reviewer's comment regarding the glucose analog 2-DG, which is widely used to inhibit glycolysis. Notably, recent studies have used it to show that glycolytic activity is critical for reactivating HIV-1 in macrophage reservoirs (Real et al., 2022, PMID 36220814). In our study, we did not initially use 2-DG because it also inhibits glucose contribution to OXPHOS, making it challenging to distinguish between the roles of glycolysis and OXPHOS in macrophages (Wang et al., Cell Metabolism, PMID 30184486). Unlike Oxamate or GSK 2837, which specifically target LDHA, 2-DG does not exclusively affect glycolysis. To address the reviewer's comment, we conducted the suggested experiments using 2-DG in CmMTB-treated macrophages during the revision process. The results are provided in Figure S3D and S3G-H) and commented on page 7, line 38: "Of note, blocking glucose uptake with 2-DG completely abrogates HIV-1-infected cell fusion into MGCs (Figure 3D and S3G-H)". We also discussed the results obtained with this drug on page 11, Lines 35-39.

A standard glycolytic function test (glucose, oligomycin and 2-DG injection) should be performed to assess the effect of TB-PE and cmMTB on the macrophages directly.

We appreciate the reviewer's comment and have addressed it by performing new experiments to test CmMTB's ability to alter macrophage glycolytic activity using the Seahorse Glycolytic Rate Assay. This assay, a refined version of the classical Seahorse Glycolysis Rate Test (see <https://www.agilent.com/en/products/cell-analysis/glycolysis-assays-using-cell-analysis-technology>), relies on an algorithm that generates the Proton Efflux Rate (PER), providing a robust quantitative measurement of glycolytic function. The new results obtained for 5 donors have been included in Figure 2A, and commented in the results section (Page 6, lines 30-33). The results confirmed that "all the parameters related to glycolysis (basal, % of Proton Efflux Rate (PER) and compensatory) were significantly increased in cmMTB-differentiated cells compared to controls (Figure 2A)."

Depriving glucose is not the best way to show the effect of glucose on HIV infection and MGC formation, as it can affect other aspects of cellular physiology, such as redox and bioenergetics. Instead, the use of galactose in place of glucose would generate ATP only by OXPHOS. Some key experiments should be repeated using galactose as a sole C source.

We agree with this comment. In M2 macrophages, both glucose deprivation (as demonstrated in this study, Figure 3H-K) and glucose substitution with galactose (Wang et al., Cell Metabolism, PMID 30184486) effectively suppress glycolytic activity. Galactose must first be metabolized via the Leloir pathway before entering glycolysis, thereby reducing glycolytic flux. As suggested by the reviewer, we conducted new experiments to complement our study, using galactose as the carbon source instead of glucose during the revision process. As shown in the new Figure S3K-L, culturing cmMTB-differentiated macrophages in galactose instead of glucose fails to rescue HIV-1 infection levels or MGC formation,

despite preserved mitochondrial respiration. These results indicate that ATP generation through OXPHOS alone is insufficient to support TNT formation and HIV-1 cell-to-cell spread, reinforcing the conclusion that active glycolytic flux, rather than cellular energetic status per se, is specifically required for these processes.

UK5099 and oxamate nuclei seem smaller and less bright compared to the control. Images between control and UK5099 appear marginally different (non-significant).

Figure 3A may not clearly convey that the nuclei are unaffected by the treatment. To address this, we have adjusted the images, particularly the DAPI staining settings, to ensure accurate interpretation. Please see the new Figure 3A. Regarding the slight effect of UK5099 treatment on Gag expression (infection index), as discussed above, this effect is reproducible and significant. The image in Figure 3A for the UK condition has been replaced with a more representative one.

The overall impact of the study is limited as the authors provide no evidence on the mechanism of how glycolysis induces TNT formation, which needs to be more characterized.

We fully agree that understanding how glycolysis induces tunneling nanotubes (TNTs) is a crucial and challenging question. This challenge stems from the incomplete understanding of the molecular mechanisms underlying TNT formation and the contradictory results reported across different cell types.

In our study, we demonstrated that inhibiting glycolysis—using Oxamate, GSK, 2DG, or glucose deprivation +/- galactose—reduces TNT formation, whereas promoting glycolysis with UK5099 enhances their formation. We discuss in the manuscript that glycolysis likely provides the energy required for actin cytoskeletal rearrangements, which are essential for TNT formation (page 12, line 40-44). Indeed, ATP plays a critical role in supporting cellular functions depending on actin remodeling, such as cell migration and the epithelial-to-mesenchymal transition (DeWane et al., 2021, PMID33558441).

To try to investigate the molecular mechanisms underlying TNT formation in our model, we have performed several new experiments during the revision process:

- GLUT-1 localization and TNT formation: To establish a connection between glycolysis and TNT formation, we assessed the localization of the primary glucose transporter GLUT-1 in relation to TNTs in CmMTB-treated macrophages. This approach builds on previous work on microvilli, which are F-actin structures with similarities to TNTs (Hexige et al., 2015, PMID: 25561062). Confocal microscopy images show that GLUT-1 accumulates at the tip of TNT in macrophages (new Figure 4C). These results suggest that glucose uptake at the tip of TNTs may be necessary for their extension; see discussion page 12, line 29-32.
- Effect of HIF1- α inhibition on TNT formation: In a new set of experiments using siRNA-mediated HIF1- α silencing, TNT formation was quantified. As shown in Figure 5D, inhibiting HIF1- α expression reduces the formation of TNT in HIV-1-infected CmMTB-treated cells.

Based on these experiments, we believe we have provided deeper insights into the role of glycolysis in TNT formation.

Minor comments:

The manuscript does not clearly show how the total ATP was calculated from the ATP rate assay.

The method for calculating total ATP is now explicitly described in the revised manuscript's Methods section, page 19.

In figure 1 (and everywhere else) the units on the y-axis should be corrected to [pmol/min] instead of pmol and the Seahorse profiles should mention whether the axis represents OCR or ECAR.

The reviewer is correct. The axes in the relevant figures for ATP rate results have been changed (see Figure 1A, B, C, D, and Figure S1B-d).

The authors have called the macrophages highly glycolytic in first set of results which is misleading. Although the glycoATP contribution is increasing, overall ATP production is still majorly through oxidative phosphorylation (70% vs 25%).

We fully agree with the reviewer's comment. As mentioned in the Result section, "Approximately 90% of ATP production in macrophages differentiated with cmCTR came from OXPHOS; this parameter was reduced to 70% when conditioned with cmMTB (Figure 1E-F)." CmMTB and TB-PE drive macrophages toward an M2/M(IL-10) phenotype (Lastrucci et al. 2015), and based on the extensive literature on metabolism of anti-inflammatory M2 macrophages, this phenotype primarily relies on OXPHOS and fatty acid oxidation (for review see Biswas and Mantovani, Cell Metabolism, 2012). It is therefore logical that overall ATP production in these cells remains predominantly through OXPHOS. However, we observe a significant decrease in OXPHOS activity following CmMTB treatment, alongside a marked increase in glycolysis (Figure 1).

Referring to CmMTB-treated macrophages as 'highly glycolytic' was, indeed, inaccurate, and this terminology has been corrected (see page 5, line 9).

Fig 3: Why does the HIV gag protein signal appear as irregular large spots?

In Figure 3A, the resolution used is sufficient to quantify the number of cells positive for HIV Gag (and thus the infection index). However, it does not allow for a detailed examination of the intracellular localization of Gag as "spots". The reviewer is correct that, within macrophages, the Gag signal often appears as large and intense cytoplasmic "spots" corresponding to the VCC, as illustrated in Reviewer Figure R1. These results have been previously published by us (Souriant et al., 2019; Mascarau et al, 2021) and others.

Figure R1: VCC formation in multinucleated HIV-1-infected macrophages. Human macrophages were infected with HIV-1 (NLAd8-VSVG, 3 days) and stained with HIV-gag and CD81 to stain the VCC.

Referees cross-commenting:

I agree with the reviewer# 1 assessment. However, I feel that mechanistically paper could be improved and by performing more experiments.

We fully agree that additional experiments were essential to improve the manuscript. We have addressed all comments and performed the experiments suggested by Reviewer 2. In particular, in the revised manuscript, we have better characterized the metabolic state of our cells, provided the strongest evidence for the role of glycolysis in HIV-1 exacerbation, and further elucidated the mechanism by which glycolysis induces TNT formation.

Significance

The knowledge generated is important and valuable for future studies to understand the molecular players in regulating immunometabolism during HIV-TB coinfection.

We appreciate this positive comment.

February 3, 2026

RE: Life Science Alliance Manuscript #LSA-2025-03333-TR

Dr. Geanncarlo Lugo-Villarino
Institut de Pharmacologie et de Biologie Structurale
Tuberculosis and Infection Biology
205 Route De Narbonne
Toulouse, Haute-Garonne 31077
France

Dear Dr. Lugo-Villarino,

Thank you for submitting your revised manuscript entitled "Glycolysis inhibition in tuberculosis-driven metabolic rewiring reduces HIV-1 spread in macrophages". As you will see, Reviewer 2 is now satisfied with no further requests. In view of their support, we would be happy to publish your paper in Life Science Alliance pending final revisions necessary to meet our formatting guidelines.

MANUSCRIPT ORGANIZATION AND FORMATTING:

To avoid unnecessary delays in the acceptance and publication of your paper, please read the following information carefully. Full guidelines are available on our Instructions for Authors page, <https://www.life-science-alliance.org/authors>

- Please add your main, supplementary figure, table, and movie legends to the main manuscript text after the references section and remove the separate supporting information file.
- We encourage you to revise the legend for Figure S4 such that the figure panels are introduced in alphabetical order.
- Please upload your Tables in editable .doc or Excel format; Supplementary Tables should be numbered consecutively with Arabic numerals (S1, S2, S3, S4); They can be uploaded as separate files.
- Please advise your Secondary Corresponding Author to add their ORCID ID - they should have received instructions on how to do so.
- Please add the X and Bluesky handles of your host institute/organization, as well as your own and/or one of the authors, in our system.
- The titles in both the system and the manuscript file must be consistent with each other.
- Please be sure that the authorship listing and order are correct and match between the system and the manuscript file.
- Please remove the eTOC summary from the title page.
- Please rename "DISCLOSURE OF CONFLICTS OF INTEREST" to "Conflict of Interest".
- In the methods section under Live Imaging, please indicate the stage temperature.
- Please add a "Data Availability" section, placed after the Materials & Methods section. This section should refer to the sequencing dataset currently under the Transcriptomics and GSEA analysis section. Please consult our guidelines at <https://www.life-science-alliance.org/manuscript-prep#format>
- Please add an Author Contributions section to your main manuscript text
- Please upload your Figure 6, i.e., Graphical Abstract, with file designation "Graphical Abstract" and remove its legend and callouts. The Graphical Abstract will appear online with the article, but will not be included in the pdf file associated with your paper.
- Please add callouts for Figure S4E and F to your main manuscript text.
- Please indicate the molecular weight for HIF1a in the capillary western shown in Supplemental Figure 5C.
- The meaning of the term "exacerbation" in the abstract (line 43) is not clear in this context. Please amend the wording with clarity for the reader in mind. Similarly, the phrase "partially mediates" in the following line is not clear, although we appreciate this change was made at the request of reviewers. We suggest changing this to "contributes to".

If you are planning a press release on your work, please inform us immediately to allow informing our production team and

scheduling a release date.

LSA encourages authors to provide a 30-60 second video where the study is briefly explained. We will use these videos on social media to promote the published paper and the presenting author (for examples, see <https://docs.google.com/document/d/1-UWCfbE4pGcDdcgzcmiuJI2XMBJnxKYeqRvLLrLSo8s/edit?usp=sharing>). Corresponding or first-authors are welcome to submit the video. Please submit only one video per manuscript. The video can be emailed to contact@life-science-alliance.org

FINAL FILES:

The following items are required for acceptance.

The license to publish form must be signed before your manuscript can be sent to production. A link to the license to publish form will be available to the corresponding author only. Please take a moment to check your funder requirements.

Thank you for your attention to these final processing requirements. Please revise and format the manuscript and upload materials as soon as you are able.

Thank you for this interesting contribution to the literature. We look forward to publishing your paper in Life Science Alliance.

Sincerely,

Reviewer #2 (Comments to the Authors (Required)):

The authors have addressed all my previous comments. I have no further concerns; the manuscript adds significant knowledge in terms of how metabolic state regulates HIV- infection.

February 6, 2026

RE: Life Science Alliance Manuscript #LSA-2025-03333-TRR

Dr. Geanncarlo Lugo-Villarino
Institut de Pharmacologie et de Biologie Structurale
Tuberculosis and Infection Biology
205 Route De Narbonne
Toulouse, Haute-Garonne 31077
France

Dear Dr. Lugo-Villarino,

Thank you for submitting your Research Article entitled "Glycolysis inhibition in tuberculosis-driven metabolic rewiring reduces HIV-1 spread in macrophages". It is a pleasure to let you know that your manuscript is now accepted for publication in Life Science Alliance. Congratulations on this interesting work.

Your manuscript will now progress through copyediting and proofing. It is journal policy that authors provide original data upon request. During proofing please ensure that the manuscript title reflects the current title in our system: "Glycolysis inhibition in tuberculosis-driven metabolic rewiring reduces HIV-1 spread in macrophages".

DISTRIBUTION OF MATERIALS:

Again, congratulations on a very nice paper. I hope you found the review process to be constructive and are pleased with how the manuscript was handled editorially. We look forward to future exciting submissions from your lab.

Sincerely,
